# A deep learning framework for automated and generalized synaptic event analysis

**Philipp S O'Neill[1,2,3], Martín Baccino-Calace[1], Peter Rupprecht[2,4], Sungmoo Lee[5], Yukun A Hao[5], Michael Z Lin[5,6], Rainer W Friedrich[7,8], Martin Mueller[1,2,9], Igor Delvendahl[1,2,3]\***

[1]Department of Molecular Life Sciences, University of Zurich (UZH), Zurich, Switzerland; [2]Neuroscience Center Zurich, Zurich, Switzerland; [3]Institute of Physiology, Faculty of Medicine, University of Freiburg, Freiburg, Germany; [4]Brain Research Institute, University of Zurich, Zurich, Switzerland; [5]Department of Neurobiology, Stanford University, Stanford, United States; [6]Department of Bioengineering, Stanford University, Stanford, United States; [7]Friedrich Miescher Institute for Biomedical Research, Basel, Switzerland; [8]Faculty of Natural Sciences, University of Basel, Basel, Switzerland; [9]University Research Priority Program (URPP), Adaptive Brain Circuits in Development and Learning (AdaBD), University of Zurich, Zurich, Switzerland

**\*For correspondence:**
igor.delvendahl@physiologie.uni-freiburg.de

**Competing interest:** The authors declare that no competing interests exist.

## eLife Assessment

This paper presents miniML, an AI-based framework for the detection of synaptic events. Benchmark results presented in the paper are **compelling**, demonstrating the superiority of miniML over current state-of-the-art alternatives. The performance of miniML is demonstrated across various experimental paradigms, showing that miniML has the potential to become a **valuable** tool for the analysis of synaptic signals.

**Abstract** Quantitative information about synaptic transmission is key to our understanding of neural function. Spontaneously occurring synaptic events carry fundamental information about synaptic function and plasticity. However, their stochastic nature and low signal-to-noise ratio present major challenges for the reliable and consistent analysis. Here, we introduce miniML, a supervised deep learning-based method for accurate classification and automated detection of spontaneous synaptic events. Comparative analysis using simulated ground-truth data shows that miniML outperforms existing event analysis methods in terms of both precision and recall. miniML enables precise detection and quantification of synaptic events in electrophysiological recordings. We demonstrate that the deep learning approach generalizes easily to diverse synaptic preparations, different electrophysiological and optical recording techniques, and across animal species. miniML provides not only a comprehensive and robust framework for automated, reliable, and standardized analysis of synaptic events, but also opens new avenues for high-throughput investigations of neural function and dysfunction.

## Introduction

Synaptic communication serves as the fundamental basis for a wide spectrum of brain functions, from computation and sensory integration to learning and memory. Synaptic transmission either arises from spontaneous or action potential-evoked fusion of neurotransmitter-filled synaptic vesicles (*Kaeser and Regehr, 2014*) resulting in an electrical response in the postsynaptic cell. Such synaptic events are a salient feature of all neural circuits and can be recorded using electrophysiological or imaging techniques.

Random fluctuations in the release machinery or intracellular $Ca^{2+}$ concentration cause spontaneous fusions of single vesicles ('miniature events') (*Kavalali, 2015*), which play an important role in synaptic development and stability (*McKinney et al., 1999*; *Banerjee et al., 2021*; *Kaeser and Regehr, 2014*; *Kavalali, 2015*). Measurements of amplitude, kinetics, and timing of these events provide essential information about the function of individual synapses and neural circuits. Miniature events are therefore key to our understanding of fundamental processes, such as synaptic plasticity or synaptic computation that support neural function (*Abbott and Regehr, 2004*; *Holler et al., 2021*). For example, amplitude changes of miniature events are a proxy of neurotransmitter receptor modulation, which is thought to be the predominant mechanism driving activity-dependent long-term alterations in synaptic strength (*Huganir and Nicoll, 2013*; *Malinow and Malenka, 2002*) and homeostatic synaptic plasticity (*Turrigiano et al., 1998*; *O'Brien et al., 1998*). In addition to spontaneous vesicle fusions, synaptic events can also result from presynaptic action potentials in neural networks. Alterations in spontaneous neurotransmission have been observed in models of different neurodevelopmental and neurodegenerative disorders (*Alten et al., 2021*; *Ardiles et al., 2012*; *Miller et al., 2014*). Thus, a comprehensive analysis of synaptic events is paramount for studying synaptic function and understanding neural diseases. However, the detection and quantification of synaptic events in electrophysiological or fluorescence recordings remains a major challenge. Synaptic events are often small in size, resulting in a low signal-to-noise ratio (SNR), and their stochastic occurrence further complicates reliable detection and evaluation.

Several techniques have been developed for synaptic event detection: Finite-difference approaches use crossings of a predefined threshold, typically in either raw data (*Kim et al., 2021*), baseline-normalized data (*Kudoh and Taguchi, 2002*), or their derivatives (*Ankri et al., 1994*). Template-based methods use a predefined template and generate a matched filter via a scaling factor (*Jonas et al., 1993*; *Clements and Bekkers, 1997*), deconvolution (*Pernía-Andrade et al., 2012*) or an optimal filtering approach (*Shi et al., 2010*; *Zhang et al., 2021*). In addition, Bayesian inference (*Merel et al., 2016*), machine learning (*Wang et al., 2024*), and peak finding routines (*Mori et al., 2024*) can be used to detect synaptic events. These techniques have facilitated the analysis of synaptic events. However, they also have several relevant limitations, such as a strong dependence of detection performance on a threshold and other decisive hyperparameters, or the need for visual inspection of results by an experienced investigator to avoid false positives. The widespread use of synaptic event recordings and the difficulty in obtaining results that are reliable across investigators and laboratories highlight the need for an automated, accurate, efficient, and reproducible synaptic event analysis method.

Artificial intelligence (AI) technologies such as deep learning (*LeCun et al., 2015*) can significantly enhance biological data analysis (*Richards et al., 2022*) and thus contribute to a better understanding of neural function. Convolutional neural networks (CNNs) are especially effective for image classification, but can also be used for one-dimensional data (*Wang et al., 2016*; *Ismail Fawaz et al., 2019*). CNNs have been successfully applied in neuroscience to segment brain regions (*Iqbal et al., 2019*), detect synaptic vesicles in electron microscopy images (*Imbrosci et al., 2022*), identify spikes in $Ca^{2+}$ imaging data (*Rupprecht et al., 2021*), and localize neurons in brain slices (*Yip et al., 2021*), or neurons with fluorescence signals in time-series data (*Denis et al., 2020*; *Sità et al., 2022*).

Here, we present *miniML*, a novel approach for detecting and analyzing spontaneous synaptic events using supervised deep learning. The miniML model is an end-to-end classifier trained on an extensive dataset of annotated synaptic events. When applied to time-series data, miniML provides high-performance event detection with negligible false-positive rates, outperforming existing methods. The method is fast, robust to threshold choice, and generalizable across diverse data types, allowing for efficient and reproducible analysis, even for large datasets. We anticipate that AI-based synaptic event analysis will greatly enhance the investigation of synaptic function, plasticity, and computation from the individual synapse to the network level.

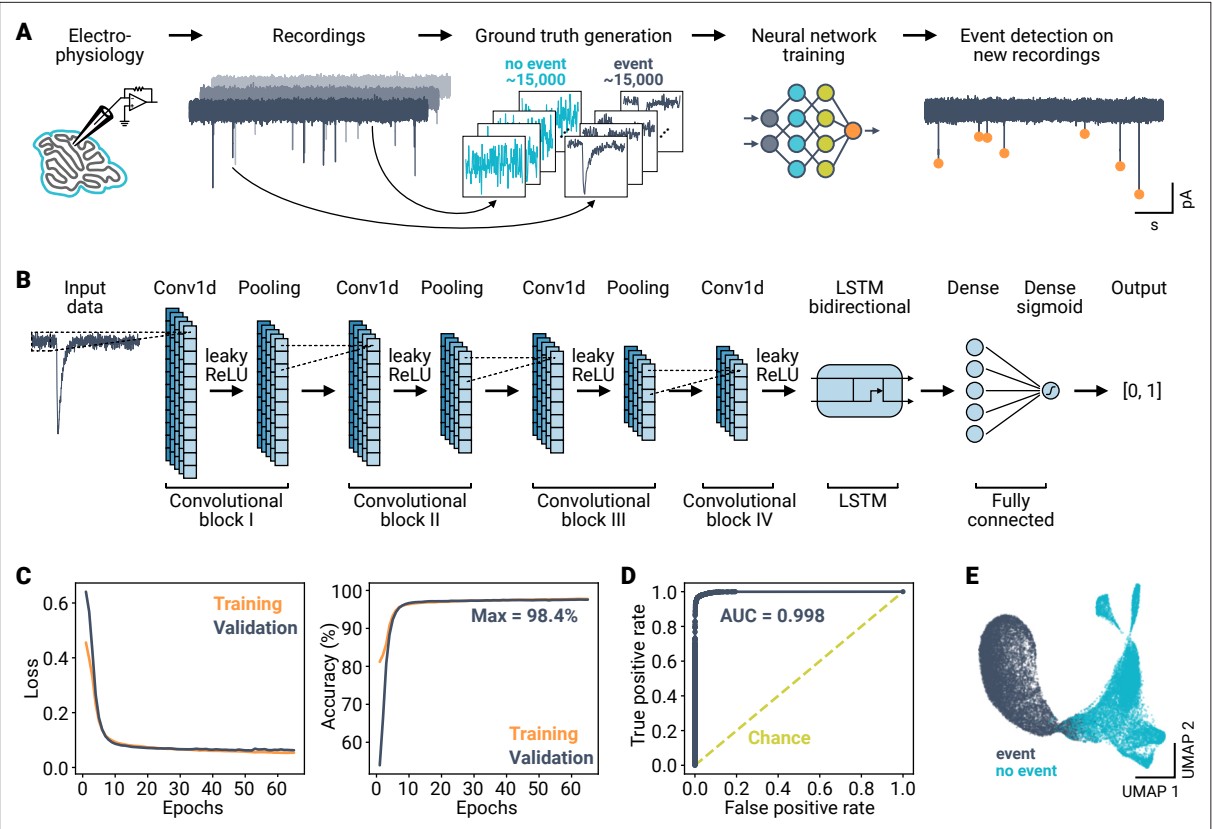

**Figure 1.** High performance classification of synaptic events using a deep neural network. (**A**) Overview of the analysis workflow. Data segments from electrophysiological recordings are extracted and labeled to train an artificial neural network. The deep learning-based model is then applied to detect events in novel time-series data. (**B**) Schematic of the model design. Data is input to a convolutional network consisting of blocks of 1D convolutional, ReLU, and average pooling layers. The output of the convolutional layers is processed by a bidirectional LSTM block, followed by two fully connected layers. The final output is a label in the interval [0, 1]. (**C**) Loss (binary crossentropy) and accuracy of the model over training epochs for training and validation data. (**D**) Receiver operating characteristic of the best performing model. Area under the curve (AUC) is indicated; dashed line indicates performance of a random classifier. (**E**) UMAP representation of the training data as input to the final layer of the model, indicating linear separability of the two event classes after model training.

The online version of this article includes the following figure supplement(s) for figure 1:

**Figure supplement 1.** Visualization of model training.

**Figure supplement 2.** Impact of dataset size, class balance, and model architecture on training performance.

## Results

### miniML enables highly accurate classification of synaptic events

To investigate whether an AI model can detect stochastic synaptic events in noisy single-trial time-series data, we designed a deep neural network consisting of CNN, long short-term memory (LSTM), and fully connected dense layers ('miniML', *Figure 1A*). The CNN-LSTM model takes a section of a univariate time-series recording as input and outputs a label for that section of data (*Figure 1B*). The miniML model is trained to classify short segments of electrophysiological data as either positive or negative for a synaptic event using supervised learning. The trained classifier can then be applied to unseen time-series data to localize events.

To train the miniML model, we first extracted a large number of synaptic events and corresponding event-free sections from previous voltage-clamp recordings of cerebellar mossy fiber to granule cell (MF–GC) miniature excitatory postsynaptic currents (mEPSCs) (*Delvendahl et al., 2019*). All samples were then visually inspected and labeled to generate the training dataset. We applied data augmentation techniques to include examples of typical false positives in the training data (Materials and methods). In total, the training data comprised ~30,000 samples that were split into training and validation sets (0.75/0.25). Across training epochs, loss decreased and accuracy increased, stabilizing

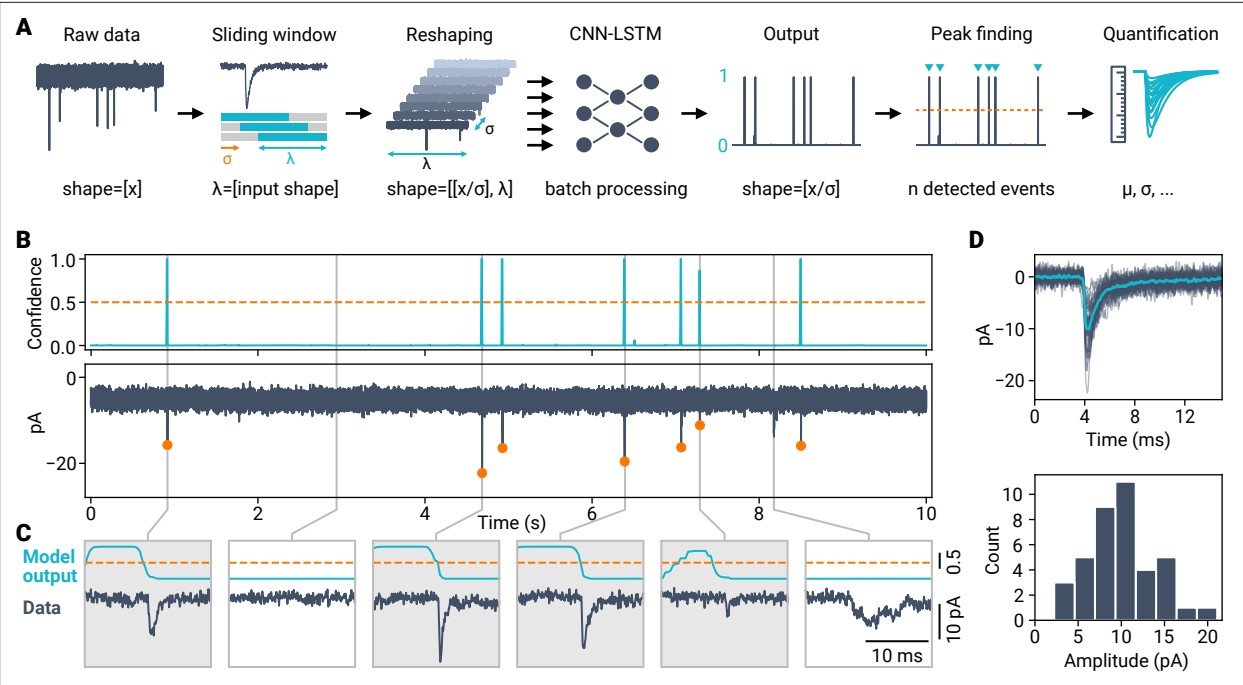

**Figure 2.** Applying AI-based classification to robustly detect synaptic events in electrophysiological time-series data. (**A**) Event detection workflow using a deep learning classifier. Time-series data are reshaped with a window size corresponding to length of the training data and a given stride. Peak detection of the model output allows event localization (orange dashed line indicates peak threshold) and subsequent quantification. (**B**) Example of event detection in a voltage-clamp recording from a cerebellar granule cell. Top: Prediction trace (top) and corresponding raw data with synaptic events (bottom). Dashed line indicates minimum peak height of 0.5, orange circles indicate event localizations. (**C**) Zoom in for different data segments. Detected events are highlighted by gray boxes. (**D**) Event analysis for the cell shown in (**B**) (total recording time, 120 s). Top: Detected events with average (light blue). Bottom: Event amplitude histogram.

The online version of this article includes the following figure supplement(s) for figure 2:

**Figure supplement 1.** Fast computation time for event detection using miniML.

**Figure supplement 2.** A graphical user interface for miniML.

**Figure supplement 3.** miniML performance on event-free data.

after ~30 epochs (*Figure 1C*). The model with the highest validation accuracy was selected for further use, achieving 98.4% (SD 0.1, fivefold cross-validation). Saliency map analysis (*Simonyan et al., 2013*) indicated that the AI model mainly relied on the data sections around the peak of synaptic events to discriminate with respect to the labels (*Figure 1—figure supplement 1*). The trained miniML model achieved an area under the receiver operating characteristic (ROC) curve close to 1 (*Figure 1D*), indicating almost perfect separability of the classes (*Figure 1E*, *Figure 1—figure supplement 1*). Deep learning typically requires large datasets for training (*van der Ploeg et al., 2014*). To investigate how miniML's classification performance depended on the dataset size, we systematically varied the number of training samples. As expected, the accuracy increased with larger datasets. However, the performance gain was marginal when exceeding 5000 samples (<0.2%, *Figure 1—figure supplement 2*), indicating that relatively small datasets suffice for effective model training (*Bailly et al., 2022*). These results demonstrate the efficacy of supervised deep learning in accurately classifying segments of neurophysiological data containing synaptic events.

## The miniML model robustly detects synaptic events in electrophysiological recordings

Synaptic events are typically recorded as continuous time-series data of either membrane voltage or current. To apply the trained classifier to detect events in arbitrarily long data, we used a sliding window approach (*Figure 2A*). Time-series data are divided into overlapping sections that correspond to the input shape of the CNN-LSTM classifier. We used a stride for the sliding window, which

reduces the number of inferences needed and speeds up the computation time while maintaining high detection performance (*Figure 2—figure supplement 1*). By reshaping the data into short sections, model inference can be run in batches and employ parallel processing techniques, including graphics processing unit (GPU) computing, resulting in analysis times of a few seconds for minute-long recordings (*Figure 2—figure supplement 1*). The miniML detection method is thus time-efficient and can be easily integrated into (high-throughput) data analysis pipelines. To facilitate the use of the method, miniML also includes a graphical user interface (*Figure 2—figure supplement 2*).

The output of the miniML model predicts the label—no event or event—for each time step (i.e. stride) with a numerical value ranging from zero to one (*Figure 1B*). These output values can be interpreted as the confidence that the corresponding data segment contains a synaptic event. Model inference thus outputs a prediction trace that has a slightly lower temporal resolution compared to the original input according to the stride of the sliding window. Using peak finding on the prediction trace allows extracting data segments with individual events from a recording. While 0.5 represents a reasonable minimum peak value, the exact choice is not critical to detection performance (see below). To quantify events, extracted sections are checked for overlapping events, and detected events are aligned by the steepest slope, allowing calculation of individual event statistics. *Figure 2B* illustrates the sliding window approach to detect spontaneous events in time-series data, such as a continuous

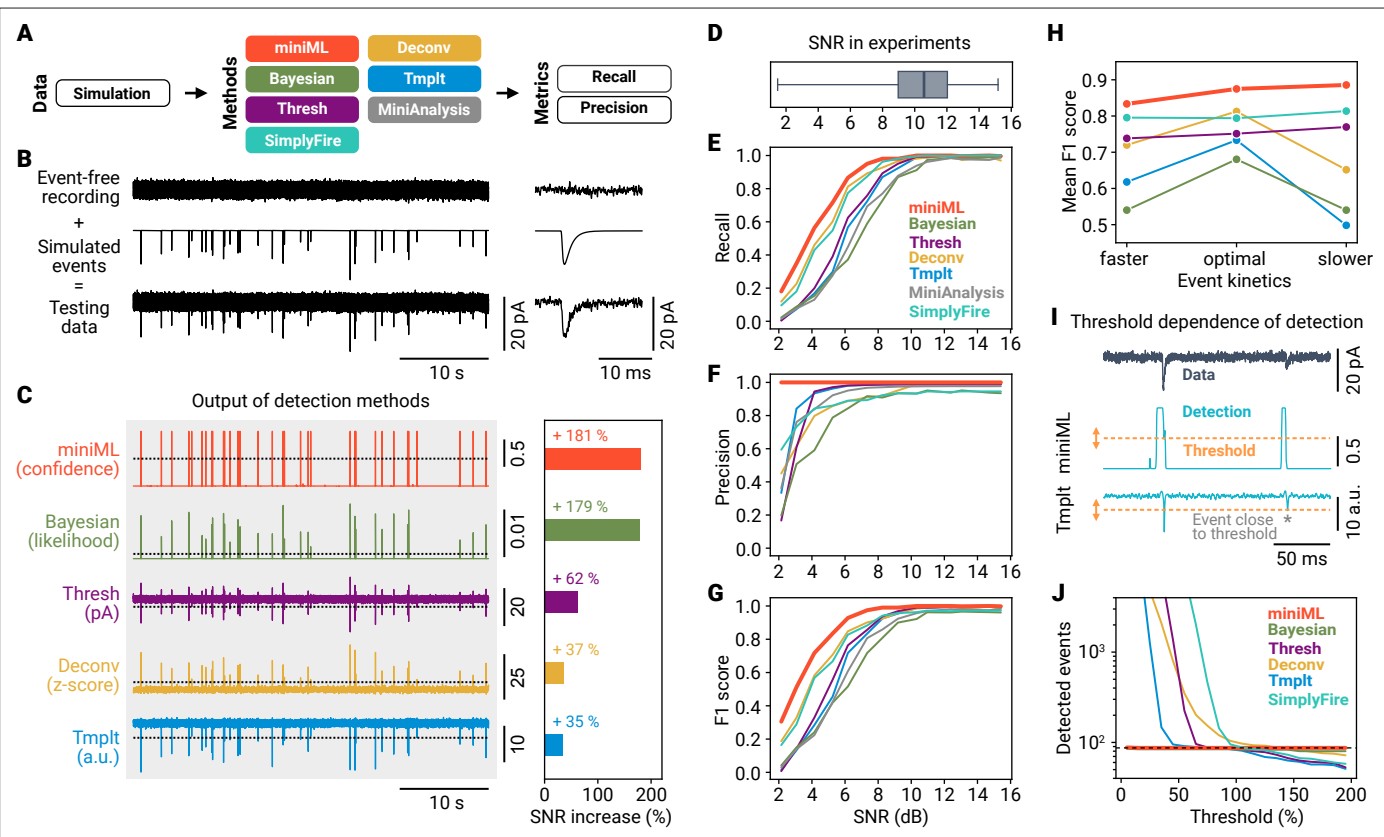

**Figure 3.** Systematic benchmarking demonstrates that AI-based event detection is superior to previous methods. (**A**) Scheme of event detection benchmarking. Six methods are compared using precision and recall metrics. (**B**) Event-free recordings were superimposed with generated events to create ground-truth data. Depicted example data have a signal-to-noise ratio (SNR) of 9 dB. (**C**) Left: Output of the detection methods for data in (**B**). Right: Improvement in SNR relative to the data. Note that MiniAnalysis is omitted from this analysis because the software does not provide output trace data. (**D**) SNR from mEPSC recordings at MF–GC synapses (n = 170, whiskers cover full range of data). (**E–G**) Recall, precision, and F1 score versus SNR for the six different methods. Data are averages of three independent runs for each SNR. (**H**) Average F1 score versus event kinetics. Detection methods relying on an event template (template-matching, deconvolution and Bayesian) are not very robust to changes in event kinetics. (**I**) Evaluating the threshold dependence of detection methods. Asterisk marks event close to detection threshold. (**J**) Number of detected events vs. threshold (in % of default threshold value, range 5–195) for different methods. Dashed line indicates true event number.

The online version of this article includes the following figure supplement(s) for figure 3:

**Figure supplement 1.** Extended benchmarking and threshold dependence of event detection.

voltage-clamp recording of spontaneous mEPSCs in a cerebellar GC. miniML provided a clear peak for all synaptic events present, without false positives (*Figure 2C*, *Figure 2—figure supplement 3*), allowing fast and reliable event quantification (*Figure 2D*). These data demonstrate that a deep learning model can be applied to detect synaptic events in electrophysiological time-series data.

## AI-based event detection is superior to previous methods

To benchmark the AI model's event detection performance, we compared it with commonly used template-based approaches (template-matching *Clements and Bekkers, 1997* and deconvolution *Pernía-Andrade et al., 2012*), and a finite-threshold-based method (*Kudoh and Taguchi, 2002*). Some of these—or similar—algorithms are implemented in proprietary software solutions to record and analyze electrophysiological data. We also included a Bayesian event detection approach (*Merel et al., 2016*), SimplyFire (*Mori et al., 2024*), and the automated event detection routine of Mini-Analysis software (Synaptosoft Inc). To compare synaptic event detection methods, we developed a standardized benchmarking pipeline (*Figure 3A*). We first performed event-free voltage-clamp recordings from mouse cerebellar GCs (in the presence of blockers of inhibitory and excitatory transmission; see Materials and methods and *Figure 2—figure supplement 3*). We then generated synthetic events with a two-exponential time course and superimposed these on the raw recording to produce ground-truth data (*Figure 3B–C*). Event amplitudes were drawn from a log-normal distribution (*Figure 3—figure supplement 1*) with varying means to cover the range of SNRs typically observed in recordings of miniature events (2–15 dB, data from n = 170 GC recordings, *Figure 3D*). We generated events with kinetics that closely matched the template used for the matched-filtering approaches, thus ensuring a conservative comparison of miniML with other methods applied under optimal conditions (i.e. using the exact average event shape as template). To measure detection performance, we calculated recall (i.e. sensitivity), precision (fraction of correct identifications) and the F1 score. Recall depended on SNR for all methods, with miniML and deconvolution showing the highest values (*Figure 3E*). The precision was highest for miniML, which detected no false positives at any SNR, in contrast to all other methods (*Figure 3F*). When assessing overall performance, miniML provided the highest F1 scores across SNRs (*Figure 3G*). In addition, miniML showed superior results when changing event kinetics, indicating higher robustness to variable event shapes (*Figure 3H*, *Figure 3—figure supplement 1*), which may be particularly important in neurons with diverse synaptic inputs due to mixed receptor populations (*Lesperance et al., 2020*), or during pharmacological experiments (*Ishii et al., 2020*).

Conventional event detection methods typically produce a detection trace with a shape identical to the input data and values in an arbitrary range (*Figure 3C*). In contrast, miniML generates output in the interval [0, 1], which can be interpreted as the confidence of event occurrence. The output of event detection methods—the detection trace—significantly increases the SNR with respect to the original data. In our benchmark scenario, miniML and the Bayesian method provided the greatest discrimination from background noise (*Figure 3C*). To locate the actual event positions, a threshold must be applied to the detection trace, which can drastically affect the results, as peaks in the detection trace may depend on the event amplitude. Intriguingly, miniML's detection trace peaks did not depend on event amplitudes, setting it apart from other methods (*Figure 3—figure supplement 1*). For template-based methods, recommendations on threshold selection are provided (*Clements and Bekkers, 1997*; *Pernía-Andrade et al., 2012*), but users usually need to adjust this parameter according to their specific data. The choice of the threshold strongly influences the detection performance, as even small changes can lead to marked increases in false positives or false negatives. To investigate the threshold dependence of different methods, we systematically varied the threshold and analyzed the number of detected events and the F1 score. Notably, miniML's detection performance remained consistent over a wide range (5–195%, *Figure 3I-J*, *Figure 3—figure supplement 1*), with false positives occurring only at the lower threshold limit (5%, corresponding to a cutoff value of 0.025 in the detection trace). Conversely, the other detection methods were very sensitive to threshold changes (*Figure 3J*, *Figure 3—figure supplement 1*) due to the comparatively low SNR ratio of the output traces (but note that the Bayesian method is only slightly threshold dependent). In addition, the miniML threshold (i.e. minimum peak) value is bounded with an intuitive meaning. These comparisons underscore that miniML requires no prior knowledge of the exact event shape and is virtually threshold independent, thus enabling reliable event detection.

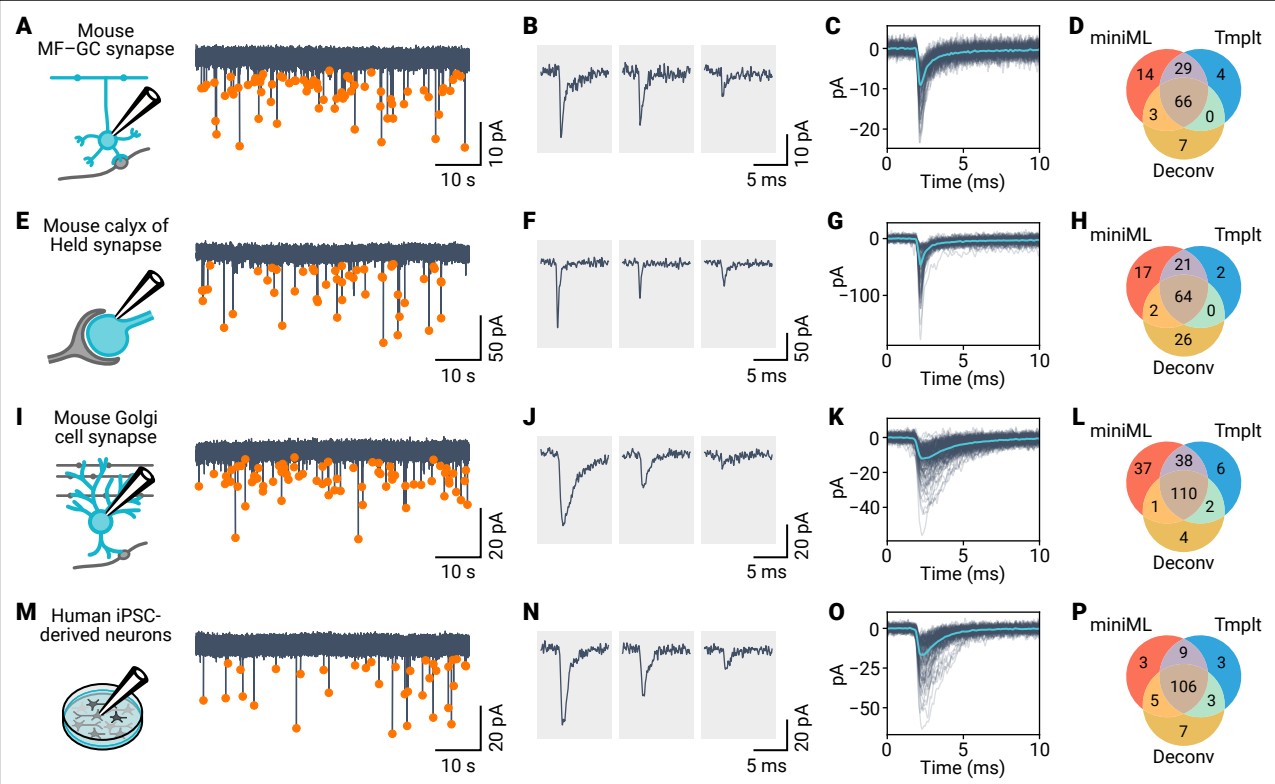

**Figure 4.** Application of miniML to electrophysiological recordings from diverse synaptic preparations. (**A**) Schematic and example voltage-clamp recordings from mouse cerebellar GCs. Orange circles mark detected events. (**B**) Representative individual mEPSC events detected by miniML. (**C**) All detected events from (**A**), aligned and overlaid with average (light blue). (**C**) Detected event numbers for miniML, template matching, and deconvolution. (**E–H**) Same as in (**A–D**), but for recordings from mouse calyx of Held synapses. (**I–L**) Same as in (**A–D**), but for recordings from mouse cerebellar Golgi cells. (**M–P**) Same as in (**A–D**), but for recordings from cultured human induced pluripotent stem cell (iPSC)-derived neurons. miniML consistently detects spontaneous events in all four preparations and retrieves more events than matched-filtering approaches.

The online version of this article includes the following figure supplement(s) for figure 4:

**Figure supplement 1.** Event detection in different synaptic preparations.

## miniML reliably detects spontaneous synaptic events in out-of-sample data

We trained the miniML model using data from cerebellar MF–GC synapses, which enabled reliable analysis of mEPSC recordings from this preparation (*Figures 2 and 4A-C*). Comparison with several previous detection methods revealed that miniML detected more events with a waveform that is consistent with mEPSCs (*Figure 4D*, *Figure 4—figure supplement 1*). Synaptic event properties, such as kinetics, SNR, or frequency, are variable between preparations. This heterogeneity is caused by, for example, differences in the number and type of postsynaptic neurotransmitter receptors, the number of synaptic inputs, postsynaptic membrane properties, or presynaptic release properties. To test miniML's generalizability, we evaluated the detection performance on out-of-sample data. We recorded and analyzed data from the mouse calyx of Held synapse (*Figure 4E*), a large axosomatic synapse in the auditory brainstem that relays rate-coded information over a large bandwidth and with high temporal precision. Due to the large number of active zones, synaptic events are quite frequent. Despite being trained on cerebellar MF–GC data, miniML accurately detected mEPSCs in these recordings (*Figure 4F–H*). This result may be facilitated by the fast event kinetics at the calyx of Held, which approach those of MF–GC mEPSCs. To test whether miniML could also detect events with slower kinetics, we applied it to recordings from cerebellar Golgi cells (*Figure 4I*; *Kita et al., 2021*). These interneurons in the cerebellar cortex provide surround inhibition to GCs and receive input from parallel fiber and MF synapses. miniML reliably detected synaptic events in these recordings (*Figure 4J–L*), although event decay kinetics were slower compared to the training data

(Golgi cell, 1.83 ms [SD 0.44 ms, n = 10 neurons], GC training data, 0.9 ms). Synaptic events are also commonly recorded from neuronal culture preparations. We determined whether miniML could be applied to recordings from cultured human induced pluripotent stem cell (iPSC)-derived neurons. We patch-clamped neurons in 8-week-old cultures of predominantly cortical glutamatergic identity (*Asadollahi et al., 2023*) and recorded spontaneous synaptic events in voltage-clamp (*Figure 4M*). Using miniML on these human iPSC-derived neuron data showed robust detection of synaptic events (*Figure 4N–P*), which had an average frequency of 0.15 Hz (SD 0.24 Hz, n = 56 neurons). Taken together, the consistent performance across different synaptic preparations indicates that the miniML model can be directly applied to out-of-sample data with similar kinetics. miniML also provided higher event detection accuracy than template-based or finite-threshold-based detection methods, which are prone to false positives (*Figure 4D, H, L, P*, *Figure 4—figure supplement 1*).

## Generalization of miniML to diverse event and data types via transfer learning

While applicable to out-of-sample data (*Figure 4*), simulations indicated that larger differences in event kinetics and waveform may ultimately hinder detection when using the MF–GC mEPSC model (*Figure 5—figure supplement 1*). Not only distinct event waveforms, but also different hardware

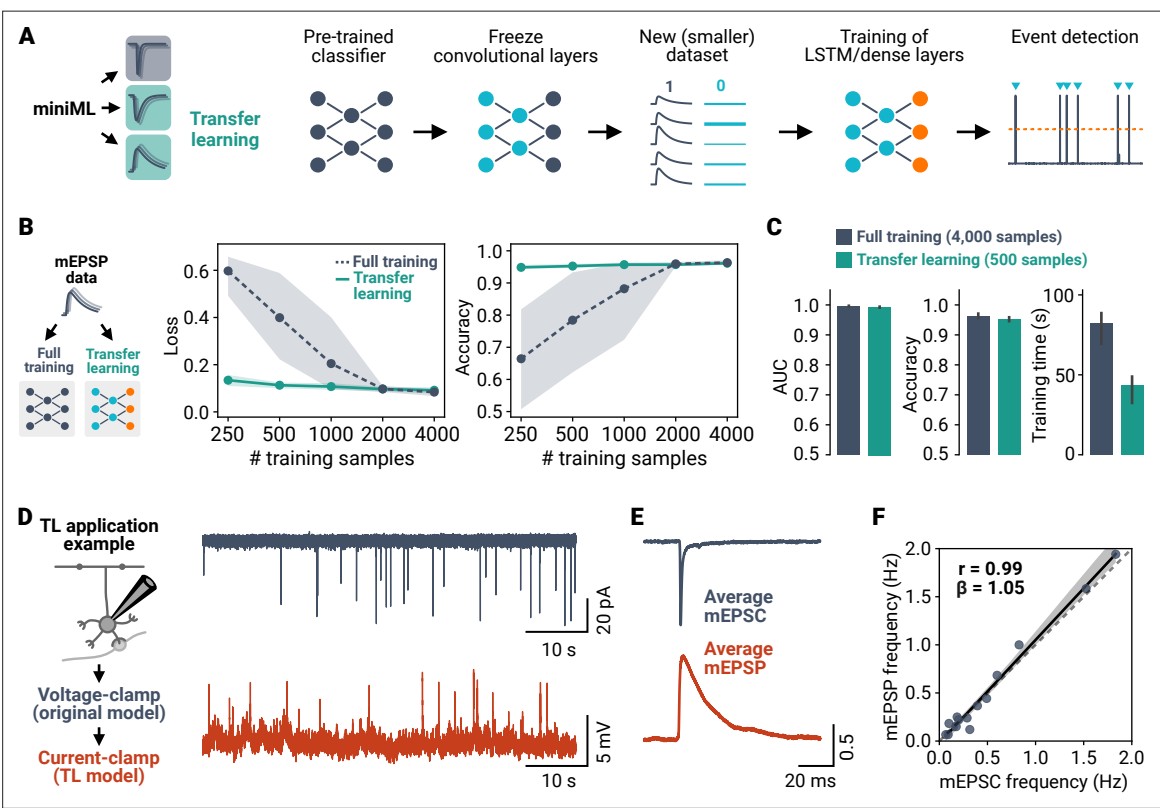

**Figure 5.** Transfer learning allows analyzing different types of events with small amounts of training data. (**A**) Illustration of the transfer learning (TL) approach. The convolutional layers of a trained miniML model are frozen before retraining with new, smaller datasets. (**B**) Comparison of TL and full training for MF–GC mEPSP data. Shown are loss and accuracy versus sample size for full training and TL. Indicated subsets of the dataset were split into training and validation sets; lines are averages of fivefold crossvalidation, shaded areas represent 95% CI. Note the log-scale of the abscissa. (**C**) Average AUC, accuracy, and training time of models trained with the full dataset or with TL on a reduced dataset. TL models yield comparable classification performance with reduced training time using only 12.5% of the samples. (**D**) Example recordings of synaptic events in voltage-clamp and current-clamp mode consecutively from the same neuron. (**E**) Average peak normalized mEPSC and mEPSP waveform from the example in (**D**). Note the different event kinetics depending on recording mode. (**F**) Event frequency for mEPSPs plotted against mEPSCs. Dashed line represents identity, solid line with shaded area represents linear regression.

The online version of this article includes the following figure supplement(s) for figure 5:

**Figure supplement 1.** Recall depends on event kinetics.

**Figure supplement 2.** Transfer learning facilitates model training across different datasets.

and recording conditions may affect the characteristics of the recorded data. Scalability and generalizability are important for robust event detection. To facilitate the application of miniML to different preparations, recording conditions, and data types, we employed a transfer learning (TL) strategy. TL is a powerful technique in machine learning that allows for the transfer of knowledge learned from one task or domain to another (*Yosinski et al., 2014*; *Caruana, 1994*). TL is widely used with CNNs to take advantage of large pre-trained models and repurpose them to solve new, unseen tasks (*Theodoris et al., 2023*). Importantly, only a part of the network needs to be trained for the novel task, which significantly reduces the number of training samples needed and speeds up training while avoiding overfitting (*Yosinski et al., 2014*). We therefore reasoned that TL based on freezing the convolutional layers during training of our pre-trained network could be used to train a new model to detect events with different shapes and/or kinetics, using a lower number of training samples (*Figure 5A*).

We tested the use of TL for miniML with recordings of miniature excitatory postsynaptic potentials (mEPSPs) in mouse cerebellar GCs. These events have opposite polarity and slower kinetics compared to the original mEPSC training data. We compared TL-based model training to full training (with all layers trainable and reinitialized weights), varying the training sample size. Whereas accuracy increased and loss decreased with the number of samples (*Figure 5B*), TL models performed well with as few as 400 samples. Under these conditions, accuracy was only slightly lower than for full training with almost 10 times the sample size (median accuracy, 95.4 vs 96.1%; *Figure 5C*). This suggests that TL can significantly reduce the sample size needed for training, minimizing the time-consuming process of event extraction and labeling. Across different datasets, TL-trained models performed comparably to those trained from scratch (*Figure 5—figure supplement 2*). Taken together, these data demonstrate that TL allows model training with an order of magnitude smaller amount of training data, making miniML easily transferable to new datasets with diverse characteristics.

To investigate how a TL-trained model performs in event detection, we analyzed data in different recording modes (voltage-clamp vs. current-clamp). The different noise conditions typically

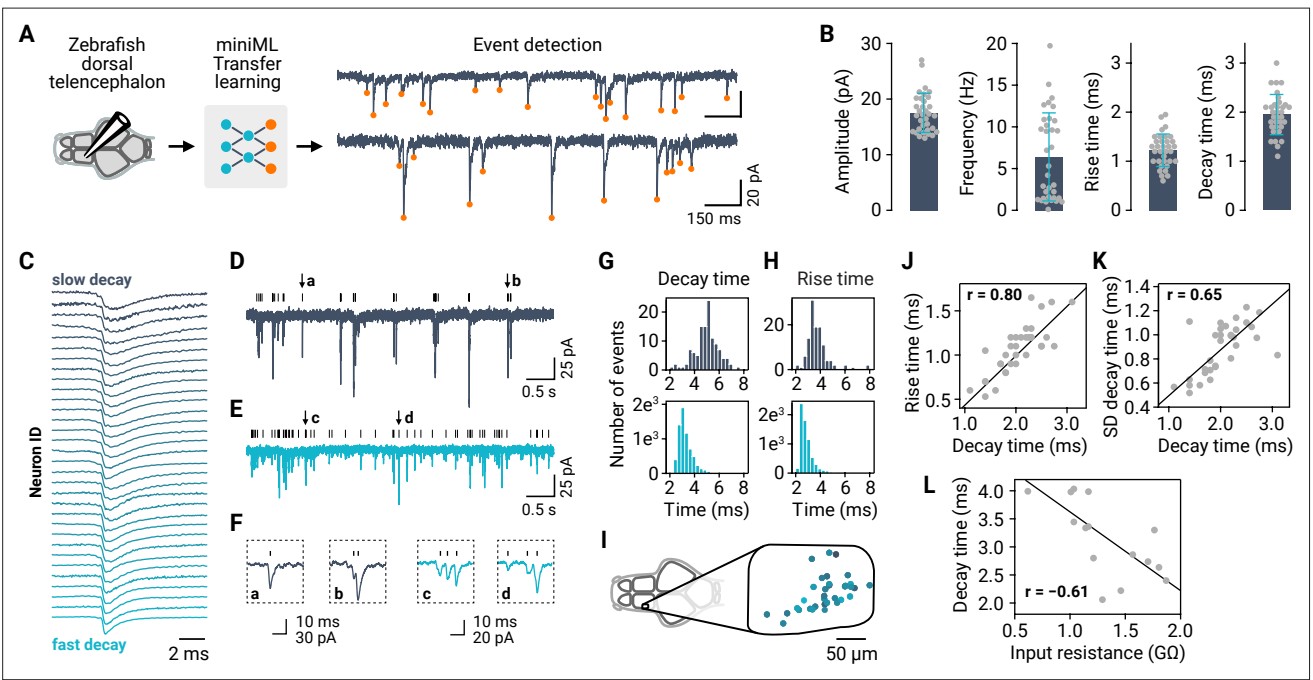

**Figure 6.** Synaptic event detection for neurons in a full-brain explant preparation of adult zebrafish. (**A**) Application of TL to facilitate event detection for EPSC recordings. (**B**) Extraction of amplitudes, event frequencies, decay times, and rise time for all neurons in the dataset (n = 34). Bars are means and error bars denote SD. (**C**) Typical (mean) event kinetics for the analyzed neurons, ordered by decay times. Event traces are peak-normalized. (**D–E**) Example recordings with slow (dark blue) and fast (light blue) event kinetics. (**F**) Examples of events taken from (**D–E**), illustrating the diversity of kinetics within and across neurons. (**G**) Distribution of event decay kinetics across single events for two example neurons (traces shown in (**D–E**)). (**H**) Distribution of event rise kinetics across single events for same two example neurons. (**I**) Mapping of decay times (color-coded as in (**C**)) onto the anatomical map of the recording subregion of the telencephalon. (**J**) Mean decay and rise times are correlated across neurons. (**K**) Decay time distributions are broader (SD of decay time distributions) when mean decay times are larger. (**L**) Input resistance as a proxy for cell size is negatively correlated with the decay time.

encountered in current-clamp and the difference between EPSP and EPSC waveforms necessitate distinct detection schemes for common detection methods. Recording miniature EPSPs and EPSCs in the same cerebellar GCs (*Figure 5D–E*) enabled us to compare detection performance via event frequency. mEPSPs could be approximated by a two-exponential time course, similar to postsynaptic currents. However, their kinetics were considerably slower due to the charging of the plasma membrane (*Figure 5E*). We used a TL-trained model for mEPSP detection and the standard miniML model for mEPSCs. Remarkably, the average event frequencies were very similar in the two different recording modes (voltage-clamp: 0.49 Hz, SD 0.53 Hz, current-clamp: 0.54 Hz, SD 0.6 Hz, n = 15 for both) and highly correlated across neurons (*Figure 5F*). This highlights the reliability of TL with small training sets for the consistent detection of synaptic events in datasets with varying characteristics.

## miniML reveals diversity of synaptic event kinetics in an ex vivo whole brain preparation

The presence of overlapping and highly variable event shapes pose additional challenges for event detection methods. To evaluate miniML's performance in such complex scenarios, we analyzed a dataset recorded from principal neurons of the adult zebrafish telencephalon (*Rupprecht and Friedrich, 2018*). We focused on spontaneous excitatory inputs to these neurons, characterized by diverse event shapes and frequencies (*Rupprecht and Friedrich, 2018*). Training via TL (see Materials and methods) yielded a model that enabled the reliable detection of spontaneous excitatory currents (*Figure 6A*). Analysis of event properties across cells revealed broad distributions of event statistics (*Figure 6B*), including a large diversity of event rise and decay kinetics (*Figure 6B–C*). Notably, miniML consistently identified synaptic events with diverse kinetics and shapes.

We next used the extracted event kinetics features of individual neurons (*Figure 6D–H*) to demonstrate miniML's utility in better understanding the diversity of an existing dataset. First, we explored whether the anatomical location of each neuron could predict event decay times. We mapped the recorded neurons to an anatomical reference and plotted decay times as a function of their position but did not find a strong relationship (*Figure 6I*; correlation with position p > 0.05 in all three dimensions). Second, we tested the hypothesis that slower event kinetics are associated with larger cells. In large cells, EPSCs may undergo stronger filtering as they propagate from synaptic sites to the soma. Consistent with this idea, we observed correlations between decay and rise times across neurons (*Figure 6J*). Furthermore, the distribution of decay times (examples shown in *Figure 6H*) was broader for neurons with longer decay times (*Figure 6K*), suggesting a broader distribution of distances from synapses to the cell body. Finally, for a subset of neurons, we recorded input resistance, which approximates the cell membrane resistance and is therefore a proxy for cell size. Input resistance was negatively correlated with decay times across neurons (*Figure 6L*), consistent with the hypothesis that diverse event kinetics across neurons are determined by the conditions of synaptic event propagation to the soma and, more specifically, cell size. Taken together, this analysis underscores the versatility of miniML, as it can be successfully applied to new datasets with varying recording conditions. miniML consistently extracted synaptic events across a spectrum of event kinetics, enabling the identification and investigation of key factors determining event kinetics and other event-related properties across neurons.

## miniML robustly detects mEPSC differences upon genetic receptor perturbation

We next applied miniML to analyze data obtained from the *Drosophila melanogaster* larval neuromuscular junction (NMJ) (*Baccino-Calace et al., 2022*). This synapse is characterized by a higher frequency of spontaneous synaptic events with a slower time course compared with typical brain slice recordings. In addition, mEPSC recordings are performed using two-electrode voltage-clamp, which can be associated with large leak currents and rather low SNR. Because of the different event shapes in these data, we used a small dataset of manually extracted NMJ events to train a TL model. The TL model was able to reliably detect synaptic events in wild-type (WT) NMJ recordings (*Figure 7A–C*). We next assessed event detection upon deletion of the non-essential glutamate receptor subunit *GluRIIA*, which causes a strong reduction in mEPSC amplitude and faster kinetics (*DiAntonio et al., 1999*). A separate TL model allowed reliable synaptic event detection in recordings from *GluRIIA* mutant larvae (*Figure 7D–F*). We observed a 54% reduction in mEPSC amplitude compared to WT

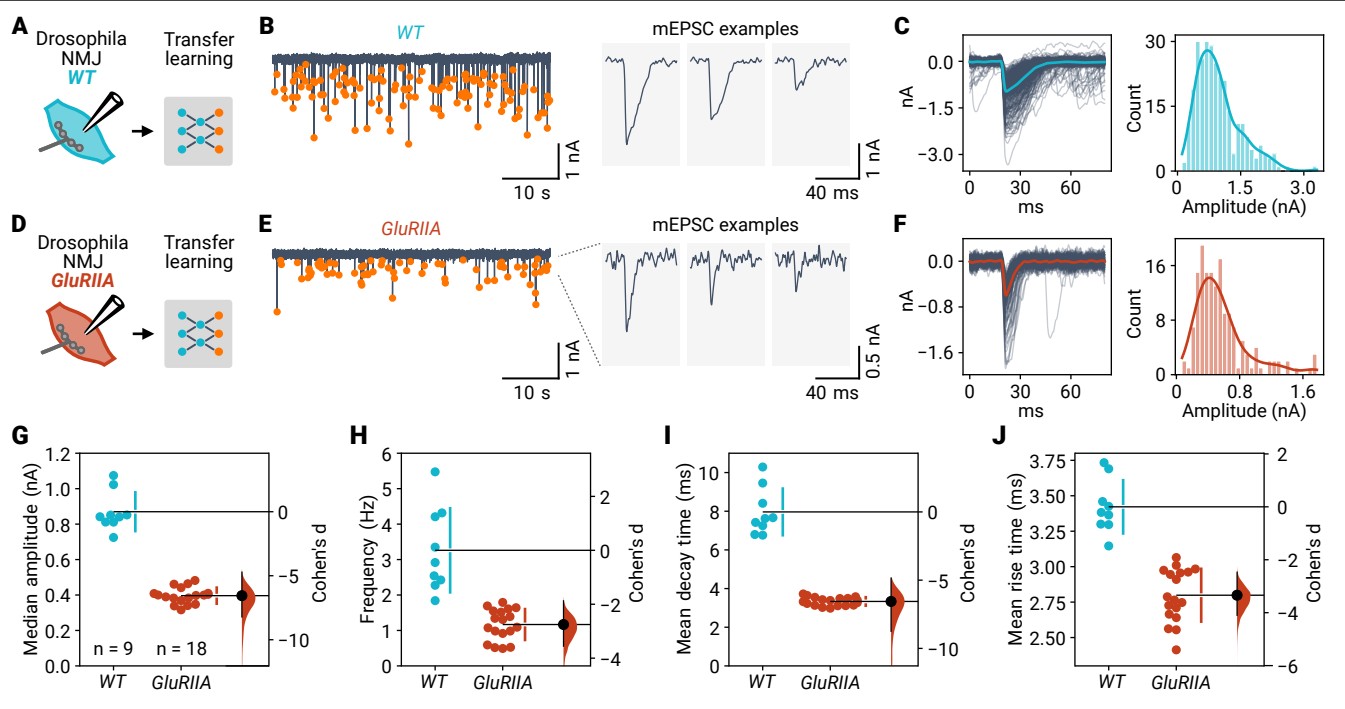

**Figure 7.** Event detection at *Drosophila* neuromuscular synapses upon altered glutamate receptor composition. (**A**) Two-electrode voltage-clamp recordings from wild-type (WT) *Drosophila* NMJs were analyzed using miniML with transfer learning. (**B**) Left: Example voltage-clamp recording with detected events highlighted. Right: Three individual mEPSCs on expanded time scale. (**C**) Left: All detected events from the example in (**B**) overlaid with the average (blue line). Right: Event amplitude histogram. (**D–F**) Same as in (**A–C**), but for *GluRIIA* mutant flies. (**G**) Comparison of median event amplitude for WT and GluRIIA NMJs. Both groups are plotted on the left axes; Cohen's d is plotted on a floating axis on the right as a bootstrap sampling distribution (red). The mean difference is depicted as a dot (black); the 95% confidence interval is indicated by the ends of the vertical error bar. (**H**) Event frequency is lower in *GluRIIA* mutant NMJs than in WT. (**I**) Knockout of *GluRIIA* speeds event decay time. (**J**) Faster event rise times in *GluRIIA* mutant NMJs.

(*Figure 7G*), consistent with previous reports (*DiAntonio et al., 1999*; *Petersen et al., 1997*). In addition, the event frequency was reduced by 64% (*Figure 7H*). Although event amplitude distributions had a similar shape in both genotypes (*Figure 7C and F*), small events below the detection limit in *GluRIIA* synapses may contribute to the observed frequency difference. Half decay and rise times were also shorter at *GluRIIA* than at WT NMJs (−58% and −18%, respectively; *Figure 7I–J*), which can be explained by the faster desensitization of the remaining *GluRIIB* receptors (*DiAntonio et al., 1999*). Thus, miniML can be applied to two-electrode voltage clamp recordings at the *Drosophila* NMJ and robustly resolves group differences upon genetic receptor perturbation.

## miniML enables reliable analysis of optical miniature events from live imaging experiments

To assess miniML's generalizability to non-electrophysiological data, we applied it to analyze events in time-series data from live fluorescence imaging experiments. Recent developments of highly sensitive fluorescent probes have enabled the recording of synaptic events using various imaging techniques (*Hao et al., 2024*; *Aggarwal et al., 2023*; *Abdelfattah et al., 2023*; *Ralowicz et al., 2024*). These technological advances offer exciting new possibilities in the study of synaptic function, but also generate novel challenges for the detection and analysis of synaptic events. Live imaging datasets typically feature a lower sampling rate, lower SNR, and distinct noise profile compared with electrophysiological recordings. Nevertheless, the waveforms of imaged synaptic release events, voltage changes, or $Ca^{2+}$ transients closely resemble those used to train miniML. Thus, we hypothesized that miniML could also be employed for event detection in fluorescence imaging data.

We first used miniML to analyze a previously published dataset *Aggarwal et al., 2023* from rat neuronal cultures expressing the glutamate sensor iGluSnFR3 (*Figure 8A–B*). These data, recorded in

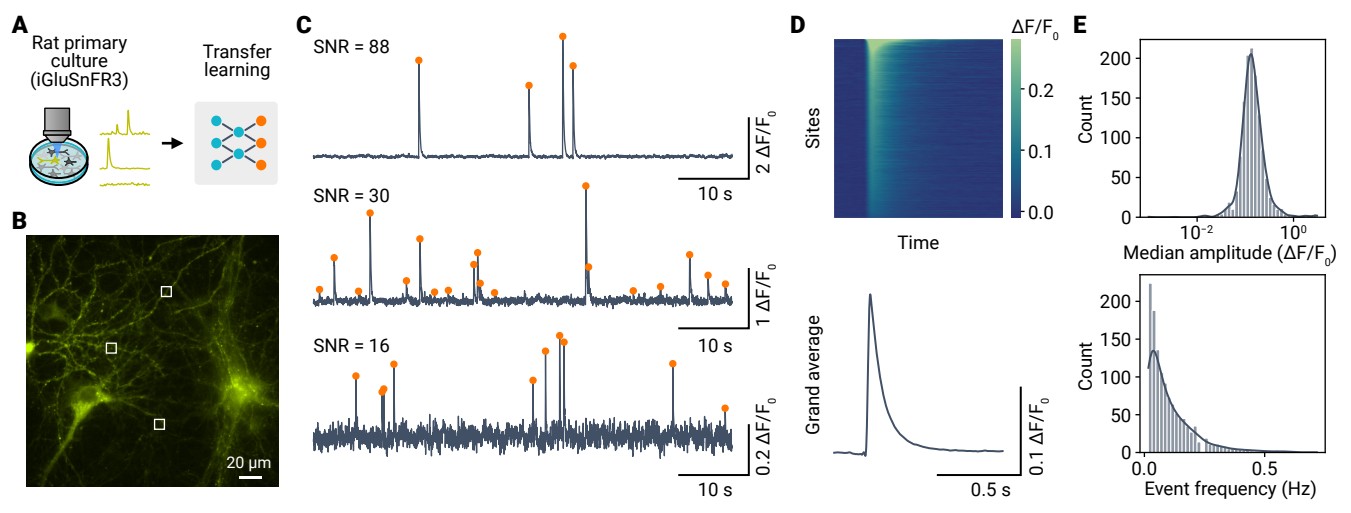

**Figure 8.** Optical detection of spontaneous glutamate release events in cultured neurons using iGluSnFR3 and miniML. (**A**) miniML was applied to recordings from rat primary culture neurons expressing iGluSnFR3. Data from *Aggarwal et al., 2023*. (**B**) Example epifluorescence image of iGluSnFR3-expressing cultured neurons. Image shows a single frame with three example regions of interest (ROIs) indicated. (**C**) $\Delta F/F_0$ traces for the regions shown in (**B**). Orange circles indicate detected optical minis. Note the different signal-to-noise ratios of the examples. (**D**) Top: Heatmap showing average optical minis for all sites with detected events, sorted by amplitude. Bottom: Grand average optical mini. (**E**) Top: Histogram with kernel density estimate (solid line) of event amplitudes for n = 1524 ROIs of the example in (**B**). Note the log-scale of the abscissa. Bottom: Event frequency histogram.

the presence of TTX, contain spontaneous transient increases in fluorescence intensity representing individual release events ('optical minis'). Initially, we selected a small subset of events from the data to train a TL model. Given the low sampling rate of the imaging (100 Hz), we upsampled the data by a factor of 10 to match the model's input shape. The TL model was subsequently applied to all detected sites (n = 1524) within the widefield recording (*Aggarwal et al., 2023*). A qualitative assessment of the imaging traces showed excellent event detection, with miniML consistently localizing the iGluSnFR3 fluorescence transients at varying SNRs (*Figure 8C*). The detected optical minis had similar kinetics to those reported in *Aggarwal et al., 2023* (10–90% rise time, median 21.8 ms; half decay time, median 48.7 ms, *Figure 8D*). In addition, analysis of event frequencies across sites revealed a power-law distribution (*Figure 8E*), consistent with fractal behavior of glutamate release (*Lowen et al., 1997*). Thus, miniML can reliably detect synaptic release events in time-series data from iGluSnFR3 recordings.

To further investigate miniML's performance in imaging data, we performed simultaneous electrophysiological and fluorescence recordings of mEPSPs in cultured rat hippocampal neurons expressing ASAP5-Kv (*Hao et al., 2024*) at physiological temperature (*Figure 9A–B*). ASAP5 allows resolving small voltage changes in the mV-range, as illustrated by the close correlation of optical and current-clamp signals (*Figure 9C*). However, event detection is more challenging in voltage imaging data due to the lower SNR compared to electrophysiological recordings (ASAP5: 2.26, SD 0.46; electrophysiology: 4.79, SD 0.74; n = 5 neurons; *Figure 9—figure supplement 1*). Photon shot noise and other sources of noise in the imaging setup not only limit the SNR, but can also lead to spurious event detection (*Wilt et al., 2013*; *Sjulson and Miesenböck, 2007*).

We trained separate miniML TL models to detect mEPSPs in optical and current-clamp data (*Figure 9C–D*). Optical detection yielded events that closely resembled mEPSPs, with slightly slower decay and rise kinetics (*Figure 9E*, *Figure 9—figure supplement 1*). Analysis of the corresponding amplitudes of mEPSPs and optically detected events confirmed that ASAP5 linearly reports subthreshold voltage changes in neurons (*Figure 9E*), in line with previous reports (*Hao et al., 2024*). To provide a more quantitative assessment of detection performance, we compared miniML to established template-based analysis methods (*Figure 9F*). The simultaneous recording of membrane voltage through both optical and electrophysiological means allowed us to establish ground truth data for evaluating detection performance. We quantified the detection of events by different methods in imaging data relative to electrophysiology, with threshold settings of the matched-filtering approaches adjusted to achieve ~90% precision. High precision of event detection is crucial for robust

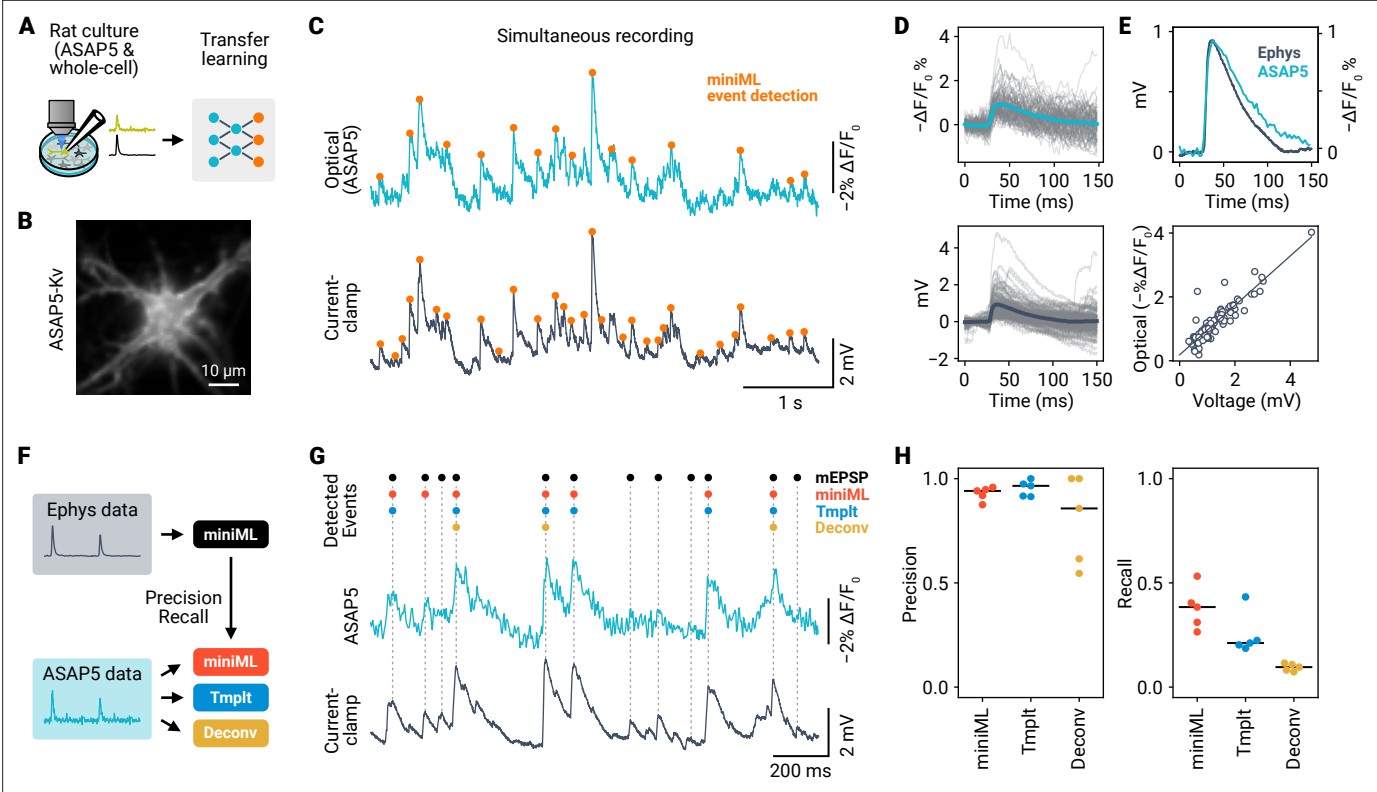

**Figure 9.** Optical detection of mEPSPs in cultured rat hippocampal neurons using ASAP5 and miniML. (**A**) Event detection in simultaneous recordings of membrane voltage using electrophysiology and ASAP5 voltage imaging. (**B**) Example epifluorescence image of an ASAP5-Kv-expressing cultured rat hippocampal neuron. (**C**) $\Delta F/F_0$ trace (Top) and simultaneous current-clamp recording (Bottom) from the neuron shown in (**B**). Events detected by miniML are indicated. (**D**) Detected events from the example recording in (**B–C**) overlaid with average (colored line). Top: ASAP5-Kv recording. Bottom: Current-clamp recording. (**E**) Top: Peak-scaled average mEPSP (dark blue) and optical events (light blue). Bottom: Amplitudes measured in optical and electrophysiological data are highly correlated (Pearson correlation coefficient $r$ = 0.9). (**F**) Comparative analysis of detection performance in ASAP5 data. (**G**) Example ASAP5 recording with detected event positions indicated for different analysis methods (miniML, template matching, deconvolution). (**H**) Precision and Recall of all three detection methods for n = 5 neurons. miniML showed highest recall (miniML, 0.38 ± 0.05; template matching, 0.25 ± 0.05 (Cohen's d, 1.25); deconvolution, 0.09 ± 0.01 (Cohen's d, 3.87); mean ± SEM) with similar precision (0.93 ± 0.01; 0.95 ± 0.02; 0.8 ± 0.1). Bars are median values.

The online version of this article includes the following figure supplement(s) for figure 9:

**Figure supplement 1.** mEPSP detection in ASAP5 recordings.

**Figure supplement 2.** Methods comparison for event detection in ASAP5 recordings.

data analysis. At comparable levels of precision, miniML demonstrated superior recall of mEPSPs compared with template matching or deconvolution (*Figure 9G–H*). Notably, miniML detected ~40% of mEPSPs in optical data, a substantial improvement over template-based methods (*Figure 9H*, *Figure 9—figure supplement 2*). These voltage imaging experiments suggest that miniML facilitates the optical detection of mEPSPs. More extensive training datasets may further increase the recall performance of miniML in voltage imaging datasets.

Together, the analysis of different optical recordings containing synaptic events demonstrates that miniML generalizes beyond electrophysiological data and can support the evaluation of fluorescence imaging datasets. miniML may thus pave the way for high-throughput analyses of spontaneous events in diverse imaging experiments.

## Discussion

Here, we developed and evaluated miniML, a novel supervised deep learning approach for detecting spontaneous synaptic events. miniML provides a comprehensive and versatile framework for analyzing synaptic events in diverse time-series data with several important advantages over current methods.

miniML outperforms existing methods for synaptic event detection, particularly with respect to false positives, which is crucial for the accurate quantification of synaptic transmission. In addition to its high precision, miniML's detection performance is robust to threshold choice. This effectively eliminates the trade-off between false positive and false negative rates typically present in event detection methods (*Merel et al., 2016*; *Clements and Bekkers, 1997*; *Pernía-Andrade et al., 2012*; *Mori et al., 2024*). Thus, miniML is highly reproducible and overcomes the need for laborious manual event inspection, enabling automated synaptic event analysis. Automated, reproducible data analysis is key to open science and the use of publicly available datasets (*Ascoli et al., 2017*; *Ferguson et al., 2014*). The Python-based software implementation of miniML enables a high degree of automatization, making it well suited for analyzing large-scale neurophysiological datasets. In addition, we provide a graphical user interface akin to MiniAnalysis or SimplyFire software.

Our results from a wide variety of preparations, covering different species and neuronal preparations, demonstrate the robustness of miniML and its general applicability to the study of synaptic physiology. miniML generalizes well to different experimental preparations, conditions, and data types. Data with event waveforms that are similar to the original miniML training data (i.e. mouse cerebellar GC recordings) can be analyzed immediately. For more distinct event waveforms or data properties, data resampling and/or retraining of a classifier via TL enables event detection optimized for the respective recording conditions. Importantly, TL requires only a few hundred labeled events for training, facilitating its application to novel datasets. For example, an existing miniML model can be easily retrained to eliminate potential false positives, for example due to sudden changes in noise characteristics. This scalability is a key advantage of the miniML approach over traditional methods.

Event detection with miniML is not restricted to electrophysiological data, but can also be applied to optical time-series data derived from live imaging experiments using, for example reporters of glutamate (*Aggarwal et al., 2023*), $Ca^{2+}$ (*Tran and Stricker, 2021*), or membrane voltage (*Li et al., 2020*). Optical recordings of subthreshold events enable high-throughput or subcellular characterizations of synaptic function in vitro or in vivo, but they usually exhibit lower SNR and lower sampling rate than electrophysiological recordings. The development of novel optical sensors with improved sensitivity and temporal resolution (*Evans et al., 2023*; *Zhang et al., 2023*; *Hao et al., 2024*) facilitates optical recordings of synaptic events, but also necessitates efficient and robust event detection methods. Our results demonstrate that miniML performs better than previous methods in identifying small synaptic events in fluorescence imaging data. The combination of advanced voltage imaging techniques and AI-based analysis has the potential to transform our ability to extract meaningful information from optical recordings at limiting SNR and may open up new possibilities for studying synaptic activity across large populations of neurons simultaneously. miniML can also be used for the analysis of evoked postsynaptic responses, such as failure analysis or quantification of unitary events, and for functional connectivity studies (*Campagnola et al., 2022*). In addition, the method could be extended to other areas of biology and medicine where signal detection is critical, such as clinical neurophysiology or imaging.

Comprehensive performance comparison is essential for evaluation and selection of analysis methods. Benchmarking requires standardized ground truth data, but unlike, for example spike inference from $Ca^{2+}$ imaging (*Theis et al., 2016*), these are usually not available for spontaneous synaptic event recordings. Furthermore, the results can vary between simulated and real data (*Theis et al., 2016*). We therefore established a benchmarking pipeline using event-free recordings with synthetic events, circumventing differences in the noise power spectrum of simulations vs. recordings (*Merel et al., 2016*; *Pernía-Andrade et al., 2012*). This approach may provide a general toolbox for evaluating the effectiveness of different synaptic event detection methods. Our comparison of detection performance in real-world data further underscores that miniML's detection performance surpasses existing methods.

The supervised learning approach of miniML requires labeled training data. Although TL requires only a few hundred training samples, these data need to be collected and annotated by the user. Future directions may explore unsupervised learning techniques such as autoencoders to reduce the dependence on annotated training data. In addition, at very high event frequencies, individual synaptic events may overlap, making their separation difficult. Although miniML can generally detect overlapping events, very close events may not always be detected and there may be a lower bound. Implementing additional techniques such as wavelet transformation, or spectral domain or shapelet

analysis (*Ye and Keogh, 2009*; *Batal and Hauskrecht, 2009*) may improve the accuracy of event detection, especially for overlapping events.

miniML presents an innovative data analysis method that will advance the field of synaptic physiology. Its open-source Python software design ensures seamless integration into existing data analysis pipelines and enables widespread use of the method, fostering the development of new applications and further innovation. Remarkably, despite its deep learning approach, miniML runs at relatively rapid speed on commodity hardware. Due to its robust, generalizable, and unbiased detection performance, miniML allows researchers to perform more accurate and efficient synaptic event analysis. A standardized, more efficient, and reproducible analysis of synaptic events will promote important new insights into synaptic physiology and dysfunction (*Lepeta et al., 2016*; *Zoghbi and Bear, 2012*) and help improve our understanding of neural function.

# Materials and methods

**Key resources table**

| Reagent type (species) or resource | Designation | Source or reference | Identifiers | Additional information |
|---|---|---|---|---|
| Strain, strain background (*Mus musculus*, ♀ and ♂) | C57BL/6JRj | Janvier Labs | RRID:MGI:2670020 | |
| Chemical compound, drug | D-APV | Tocris | Cat. # 0106 | 50 µM |
| Chemical compound, drug | NBQX | HelloBio | Cat. # HB0443 | 10 µM |
| Chemical compound, drug | Bicuculline | Sigma | Cat. # 14343 | 10 µM |
| Chemical compound, drug | Strychnine | Sigma | Cat. # S8753 | 1 µM |
| Software, algorithm | Patchmaster | HEKA Elektronik | RRID:SCR_000034 | Version 2x90.5 |
| Software, algorithm | Mini Analysis Program | Synaptosoft | RRID:SCR_002184 | Version 6.0 |
| Software, algorithm | Python | Python Software Foundation | RRID:SCR_008394 | Version 3.9 or 3.10 |
| Software, algorithm | TensorFlow | Google | RRID:SCR_016345 | Version 2.12 or 2.15 |
| Other | Borosilicate glass | Science Products | Cat. # GB150F-10P | Outer/inner diameter: 1.5/0.86 mm |

## Electrophysiological recordings

Animals were treated according to national and institutional guidelines. All experiments were approved by the Cantonal Veterinary Office of Zurich (approval number no. ZH206/2016 and ZH009/2020). Experiments were performed in male and female C57BL/6 J mice (Janvier Labs, France, RRID:MGI:2670020). Mice were 1–5 months-old, but for recordings from the Calyx of Held, which were performed in P9 animals. Animals were housed in groups of 3–5 in standard cages under a 12h-light/12h-dark cycle with food and water ad libitum. Mice were sacrificed by rapid decapitation after isoflurane anesthesia. The cerebellar vermis was quickly removed and mounted in a chamber filled with chilled extracellular solution. Parasagittal 300-µm-thick slices were cut with a Leica VT1200S vibratome (Leica Microsystems, Germany), transferred to an incubation chamber at 35 °C for 30 min, and then stored at room temperature until experiments. The extracellular solution (artificial cerebrospinal fluid, ACSF) for slicing and storage contained (in mM): 125 NaCl, 25 NaHCO$_3$, 20 D-glucose, 2.5 KCl, 2 CaCl$_2$, 1.25 NaH$_2$PO$_4$, 1 MgCl$_2$, aerated with 95% O$_2$ and 5% CO$_2$.

Slices were visualized using an upright microscope with a 60×, 1 NA water immersion objective, infrared optics, and differential interference contrast (Scientifica, UK). The recording chamber was continuously perfused with ACSF. For event-free recordings, we blocked excitatory and inhibitory transmission using ACSF supplemented with 50 µM D-APV, 10 µM NBQX, 10 µM bicuculline, and 1 µM strychnine. Patch pipettes (open-tip resistances of 3–8 MΩ) were filled with solution containing (in mM): 150 K-D-gluconate, 10 NaCl, 10 HEPES, 3 MgATP, 0.3 NaGTP, 0.05 ethyleneglycol-bis(2-aminoethylether)-N,N,N',N'-tetraacetic acid (EGTA), pH adjusted to 7.3 with KOH. Voltage-clamp and current-clamp recordings were made using a HEKA EPC10 amplifier controlled by Patchmaster software (HEKA Elektronik GmbH, Germany, RRID:SCR_000034). Voltages were corrected for a liquid junction potential of +13 mV. Experiments were performed at room temperature (21–25 °C). Miniature EPSCs (mEPSCs) were recorded at a holding potential of −100 mV or −80 mV, and miniature EPSPs

(mEPSPs) at the resting membrane potential. Data were filtered at 2.9 kHz and digitized at 50 kHz. Synaptic event recording periods typically lasted 120 s.

Voltage-clamp recordings were performed on human iPSC-derived neurons as described in *Asadollahi et al., 2023*. We recorded spontaneous EPSCs in 8-week-old neuronal cultures with cortical glutamatergic identity at a holding potential of −80 mV and with ACSF consisting of (in mM): 135 NaCl, 10 HEPES, 10 D-glucose, 5 KCl, 2 CaCl$_2$, 1 MgCl$_2$. Synaptic events were observed in ~51% of neurons. The experimental procedures for recordings of spontaneous or miniature EPSCs in zebrafish and *Drosophila* are described in *Rupprecht and Friedrich, 2018* and *Baccino-Calace et al., 2022*, respectively.

## Combined electrophysiological and optical mEPSP recordings

Cultured rat hippocampal neurons at 17–20 days in vitro were patched and imaged simultaneously at physiological temperature (30–34 °C) following the procedure described in *Hao et al., 2024*. Briefly, for electrophysiological recordings, extracellular solution containing 145 mM NaCl, 3 mM KCl, 2 mM CaCl$_2$, 2 mM MgCl$_2$, 10 mM HEPES, and 10 mM glucose (pH adjusted to 7.4, Osmolarity 310 mOsm/kg) supplemented with TTX (1 μM) and PTX (50 μM), and a pipette solution containing 123 mM K-gluconate, 10 mM KCl, 8 mM NaCl, 1 mM MgCl$_2$, 10 mM HEPES, 1 mM EGTA, 0.1 mM CaCl$_2$, 1.5 mM MgATP, 0.2 mM Na$_4$GTP, and 4 mM glucose (pH adjusted to 7.2, Osmolarity 295–300 mOsm/kg) were used. Whole-cell recordings were performed with a Multiclamp 700B amplifier, Digidata 1440 A digitizer, and pClamp software (all from Molecular Devices). Patch pipettes (open-tip resistances of 3–5 MΩ) were produced from glass capillaries with filament (outer diameter, 1.5 mm; inner diameter, 1.0 mm; King Precision Glass). Electrophysiological recordings were digitized at 10 kHz for acquisition and then lowpass filtered with 1 kHz cut-off frequency in post-processing in Clampfit.

ASAP5-Kv was excited using a SOLIS-470C LED light source (Thorlabs) with a 482/18 nm bandpass filter (Semrock); excitation light was set to provide an irradiance of 46 mW/mm$^2$. Imaging data were acquired at 400 fps using an iXon 860 EMCCD camera (Andor – Oxford Instruments) and a 525/50 nm bandpass filter (Semrock). Neurons requiring more than ±100 pA holding current to maintain the resting membrane potential near −70 mV were excluded from the analysis.

## Training data and annotation

We used synaptic event recordings from a previous publication (*Delvendahl et al., 2019*) to generate the training dataset. mEPSCs were extracted based on low-threshold template-matching. Corresponding sections of data without events were randomly selected from the recordings. The extracted windows had a length of 600 data points. Given our sampling rate of 50 kHz, this corresponds to 12 ms. We subsequently manually scored data sections as event-containing (label = 1) or not event-containing (label = 0). The ratio of events to non-events was kept close to one to ensure efficient training. Based on empirical observations of model performance, we included relatively small amplitude events that are often missed by other methods. Similarly, including negative examples that are commonly picked up as false positives, resulted in more accurate prediction traces. We used data augmentation techniques to further improve model discrimination. We simulated waveforms of non-synaptic origin, which are occasionally encountered during recordings, and superimposed them onto noise recordings. Examples included rapid current transients that can be caused by peristaltic perfusion pump systems often used in brain slices recordings, or slow currents with a symmetric rise and decay time course. A total of 4500 segments were created and labeled as 0. To maintain the ratio of negative to positive examples, we added an equivalent number of synthetic synaptic events. The biexponential waveform described in the section *Benchmarking detection methods* was used for event simulation. The final training dataset contained 30,140 samples (21,140 from recorded traces and 9,000 simulated samples).

## Deep learning model architecture

We built a discriminative end-to-end deep learning model for one-dimensional time-series classification (*Ismail Fawaz et al., 2019*). The neural network architecture comprised multiple convolutional layers, an LSTM layer, and a fully connected layer. The approach is related to networks designed for event detection in audio (*Passricha and Aggarwal, 2019*) and image data (*Islam et al., 2020*; *Donahue et al., 2017*), or classification of genomic data (*Tasdelen and Sen, 2021*). The combination

of convolutional and recurrent neural network layers helps to improve the classification performance for time-series data. In particular, LSTM layers allow learning temporal features. An initial comparison of different types of network architectures showed superior performance of a CNN-LSTM model over CNN-Dense, Residual Neural Network (ResNet), and multi-layer perceptron (MLP) architectures (*Figure 1—figure supplement 2*).

The miniML model was built using TensorFlow, an open-source machine learning library for Python (*Abadi et al., 2015*). To optimize the model architecture, we performed a Bayesian optimization of hyperparameters. Hyperparameter ranges were chosen for the free parameters of all layers. Optimization was then performed with a maximum number of trials of 50. Models were evaluated using the validation dataset. Because a higher number of free parameters tended to increase inference times, we then empirically tuned the chosen hyperparameter combination to achieve a trade-off between number of free parameters and accuracy.

The deep learning network takes batches of one-dimensional (1D) univariate time-series data as input, which are converted into a tensor of shape (batch size, data length, 1). The data is passed to three convolutional blocks. Each block consists of a 1D convolutional layer with a leaky Rectified Linear Unit (leaky ReLU) activation function followed by an average pooling layer. To avoid overfitting, each convolutional block uses batch normalization (*Ioffe and Szegedy, 2015*). Batch normalization reduced training time by about two times and improved the accuracy of the resulting model. We added a fourth convolutional block that includes a convolutional layer, batch normalization and a leaky ReLU activation function, but no average pooling layer. The output of the final convolutional layer passes through a bidirectional recurrent layer of LSTM units. The final layers consist of a fully connected dense unit layer and a dropout layer (*Srivastava et al., 2014*), followed by a single unit with sigmoid activation. The output of the neural network is a scalar between [0, 1]. The layers and parameters used, including output shape and number of trainable parameters, are summarized in *Table 1*. The total number of trainable parameters was 190,913.

## Training and evaluation

The network was trained with Tensorflow 2.12 and Python 3.10 with CUDA 11.4. Datasets were scaled between zero and one, and split into training and validation data (0.75/0.25). The model was compiled using the Adam optimizer (*Kingma and Ba, 2014*) with AMSGrad (*Reddi et al., 2019*). We trained the classifier using a learning rate $\eta$ = 2E–5 and batch size of 128 on the training data. Training was run for maximum 100 epochs with early stopping to avoid overfitting. Validation data was used to evaluate training performance. Early stopping was applied when the validation loss did not improve for eight consecutive epochs. Typically, training lasted for 20–40 epochs. We used binary cross-entropy loss and binary accuracy as measures of performance during training. The best performing model was selected from a training run, and a receiver-operating characteristic (ROC) was calculated. We used accuracy and area under the ROC curve to evaluate training performance. To accelerate training, we used a GPU; training time for the neural network was ~8 min on a workstation with NVIDIA Tesla P100 16 GB GPU (Intel Xeon 2.2 GHz CPU, 16 GB RAM).

## Model and training visualization

To analyze the discriminative data segments, we calculated per-sample saliency maps with Smooth-Grad using the Python package *tf-keras-viz* (*Simonyan et al., 2013*). Saliency maps were smoothed with a 10-point running average filter for display purposes. To illustrate the transformation by the model, we performed dimensionality reduction of the training data by Uniform Manifold Approximation and Projection (UMAP), using either the raw dataset or the input to the final layer of the deep learning model.

## Applying the classifier for event detection

The trained miniML classifier takes sections of data with predefined length as input. To detect events in arbitrarily long recordings, a sliding window procedure is used. Time-series data from voltage-clamp or current-clamp recordings is segmented using a sliding window with stride. The resulting 2D-tensor is min-max scaled and then used as model input for inference. To overcome the potential limitation of long computation times, we used a sliding window with stride procedure. Using a stride >1 significantly reduces the inference time of the model (*Figure 2—figure supplement 1*), especially for data

**Table 1.** Overview of model architecture.

| Block | Layer | Output shape | Parameters | Settings | Values |
|---|---|---|---|---|---|
| Convolutional Block I | 1D Convolutional Layer | (None, 600, 32) | 320 | Filters | 32 |
| | | | | Kernel Size | 9 |
| | | | | Padding | 'same' |
| | Batch Normalization | (None, 600, 32) | 128 | All | default |
| | Leaky ReLU | (None, 600, 32) | 0 | $\alpha$ | 0.3 |
| | Avg. Pooling | (None, 200, 32) | 0 | Pool Size | 3 |
| | | | | Strides | 3 |
| Convolutional Block II | 1D Convolutional Layer | (None, 200, 48) | 10'800 | Filters | 48 |
| | | | | Kernel Size | 7 |
| | | | | Padding | 'same' |
| | Batch Normalization | (None, 200, 48) | 192 | All | default |
| | Leaky ReLU | (None, 200, 48) | 0 | $\alpha$ | 0.3 |
| | Avg. Pooling | (None, 100, 48) | 0 | Pool Size | 2 |
| | | | | Strides | 2 |
| Convolutional Block III | 1D Convolutional Layer | (None, 100, 64) | 15'424 | Filters | 64 |
| | | | | Kernel Size | 5 |
| | | | | Padding | 'same' |
| | Batch Normalization | (None, 100, 64) | 256 | All | default |
| | Leaky ReLU | (None, 100, 64) | 0 | $\alpha$ | 0.3 |
| | Avg. Pooling | (None, 50, 64) | 0 | Pool size | 2 |
| | | | | Strides | 2 |
| Convolutional Block IV | 1D Convolutional Layer | (None, 50, 80) | 15'440 | Filters | 80 |
| | | | | Kernel Size | 5 |
| | | | | Padding | 'same' |
| | Batch Normalization | (None, 50, 80) | 320 | All | default |
| | Leaky ReLU | (None, 50, 80) | 0 | $\alpha$ | 0.3 |
| LSTM Layer | Bidirectional LSTM | (None, 96) | 135'936 | Units | 96 |
| | | | | Dropout Rate | 0.2 |
| | | | | Merge Mode | 'sum' |
| | | | | Activation | tanh |
| Fully Connected Layers | Dense | (None, 128) | 12'416 | Units | 128 |
| | | | | Activation | Leaky ReLU |
| | Dropout | (None, 128) | 0 | Dropout Rate | 0.2 |
| | Dense | (None, 1) | 129 | Units | 1 |
| | | | | Activation | sigmoid |

with high sampling rates or long recording times. This approach results in a prediction trace with data being spaced at *sampling interval * stride*. With *stride* = 20, a 120 s recording at 50 kHz sampling rate can be analyzed in ~15 s on a laptop computer (Apple M1Pro, 16 GB RAM, *Figure 2—figure supplement 1*). To maintain temporal precision, the prediction trace is resampled to the sampling frequency of the raw data.

Events in the input data trace result in distinct peaks in the prediction trace. We applied a maximum filter to enhance post-processing of the data. Thus, by applying a threshold to the prediction trace, synaptic event positions can be detected. Our analyses indicate that the absolute threshold value is not important in the [0.05, 0.95] range (*Figure 3—figure supplement 1*). Data following a threshold crossing in the prediction trace are cut from the raw data and aligned by steepest rise. To find the point of steepest rise, a peak-search is performed in the first derivative of the short data segment. If multiple peaks are detected, any peak that has a prominence ≥0.25 relative to the largest detected prominence is treated as an additional event that is in close proximity or overlapping. The resulting 2D-array with aligned events can be further analyzed to obtain descriptive statistics on, e.g., amplitude, charge, or rise and decay kinetics of individual events.

## Benchmarking detection methods

We compared the deep learning-based miniML with the following previously described detection methods: template-matching (*Clements and Bekkers, 1997*), deconvolution (*Pernía-Andrade et al., 2012*), a finite threshold-based approach (*Kudoh and Taguchi, 2002*), the commercial MiniAnalysis software (version 6.0.7, Synaptosoft), SimplyFire (*Mori et al., 2024*), and a Bayesian inference procedure (*Merel et al., 2016*). Detection methods were implemented in Python 3.9, except for the Bayesian method, which was run using Matlab R2022a, and MiniAnalysis running as stand-alone software on Windows.

We used generated standardized data to benchmark the detection performance of different methods. Event-free recordings (see section *Electrophysiological recordings* for details) were superimposed with simulated events having a biexponential waveform:

$$I(t) = (1 - e^{(\frac{-t}{\tau_{rise}})}) * e^{(\frac{-t}{\tau_{decay}})}$$

where $I(t)$ is the current as a function of time, and $\tau_{rise}$ and $\tau_{decay}$ are the rise and decay time constants, respectively. Simulated event amplitudes were drawn from a log-normal distribution with variance 0.4. Mean amplitude was varied to generate diverse signal-to-noise ratios (SNR; mean event amplitude / standard deviation of the noise). Decay time constants were drawn from a normal distribution with mean 1.0 ms and variance 0.25 ms (*Delvendahl et al., 2019*). Generated events were randomly placed in the event-free recording with an average frequency of 0.7 Hz and a minimum spacing of 3 ms. Generated traces provided ground-truth data for the evaluation of the different methods.

To quantify detection performance of different methods over a range of signal-to-noise ratios, we calculated the number of true positives (TP), false positives (FP) and false negatives (FN). From these, the following metrics were calculated:

$$Precision = \frac{TP}{TP + FP}$$

$$Recall = \frac{TP}{TP + FN}$$

$$F1 = 2 * \frac{Precision * Recall}{Precision + Recall}$$

For all metrics, higher values indicate a better model performance. We also evaluated detection performance under non-optimal conditions (i.e., events that did not precisely match the event template or the training data). To do this, we varied the kinetics of simulated events by either increasing (mean = 4.5 ms) or decreasing (mean = 0.5 ms) the average decay time constant (*Figure 3—figure supplement 1*).

## Hyperparameter settings of detection methods

For all benchmarking conditions, threshold settings were: −4 for template-matching, $5 * SD$ of the detection trace for deconvolution, and −4 pA for the finite-threshold method. Both template-based methods used a template waveform corresponding to the average of the simulated events. For the Bayesian detection approach, we used the code provided by the authors (*Merel et al., 2016*) with the following adjustments to the default hyperparameters: minimum amplitude = 3.5, noise $\phi = [0.90; −0.52]$, rate = 0.5. We chose a cutoff of $6 * SD$ of the event time posteriors as threshold. For SimplyFire, we used the following hyperparameters: direction = −1; kernel = 300; stride = 100; minimum amplitude = 4 pA. MiniAnalysis (Synaptosoft) was run in the automatic detection mode with default settings for EPSCs and amplitude and area thresholds of −4 pA and 4, respectively. We used a minimum peak height of 0.5 and minimum peak width of 10 strides for miniML. To compare different event detection methods in real-world data *Figure 4*, we used the following detection hyperparameters: We applied miniML with a window size of 600 and minimum peak width of 5 strides. For the Golgi cell data, window size was set to 900 to account for the slower event kinetics. Detection thresholds for template matching and deconvolution thresholds were set at −3.5 and 6, respectively. We adjusted the template for each preparation based on the average waveform extracted with a generic template. MiniAnalysis was run with area and amplitude thresholds of 5; amplitude threshold was increased to −20 pA and −8 pA for Calyx of Held and Golgi cell data, respectively.

## Transfer learning

To make the miniML method applicable to data with different characteristics, such as different event kinetics, noise characteristics, or recording mode, we used a transfer learning (TL) approach (*Pratt et al., 1991*). We froze the convolutional layers of the fully trained MF–GC miniML model, resulting in a new model with only the LSTM and dense layers being trainable. Thus, the convolutional layers act as pre-trained feature detectors and much fewer training samples are required. Hyperparameters and training conditions were the same as for full-training (see *Training and evaluation*) with the following exceptions: learning rate $\eta$ = 2E–8, patience = 15, batch size = 32, dropout rate = 0.5. The training data were resampled to 600 data points to match the input shape of the original model.

To compare TL with full training, random subsets of the training data (mEPSPs in cerebellar GCs, spontaneous EPSCs in zebrafish, or mEPSCs at the *Drosophila* neuromuscular junction) with increasing size were generated and used to train models using fivefold cross-validation. We always used the same size of the validation dataset. After comparing TL with full training (*Figure 5*, *Figure 5—figure supplement 2*), we trained separate models to analyze the different datasets using subsets of the available training data.

## Quantifications

Computing times were quantified using the performance counter function of the built-in Python package *time* and are given as wall times. Statistical comparisons were made using permutation t-tests with 5000 reshuffles (*Ho et al., 2019*). Effect sizes are reported as Cohen's d or mean difference, with 95% confidence intervals obtained by bootstrapping (5000 samples; the confidence interval is bias-corrected and accelerated) (*Ho et al., 2019*).

Raw data were filtered with a Hann window for event analysis after the detection. The number of points of the filter window was adjusted based on the sampling rate (default, 20 samples). Event amplitudes were quantified as difference between detected event peaks and a short baseline window before event onset. Decay times refer to half decay times (time until signal reaches 50% of the amplitude), and rise times were quantified as 10–90% rise times (time between 10% and 90% of the amplitude). These event statistics were calculated for each individual event and averaged per cell.

## Computer code

All code was implemented in Python version 3.9 or 3.10 (RRID:SCR_008394) with the following libraries: TensorFlow (RRID:SCR_016345), SciPy (RRID:SCR_008058), NumPy (RRID:SCR_008633),

Matplotlib (RRID:SCR_008624), Pandas (RRID:SCR_018214), h5py (RRID:SCR_024812), scikit-learn (RRID:SCR_002577), pyABF, dabest (RRID:SCR_022340).

## Acknowledgements

We thank Mark D Robinson and Anu G Nair for helpful discussions.

This work received funding by the Swiss National Science Foundation (grant PZ00P3 174018 to ID, grant PZ00P3 209114 to PR, grant 310030B 152833/1 to RF), the German Research Foundation (grants DE3925/1-1 and 3925/2–1 to ID), the NIH (grant 1UM1MH136462 to MZL), the Novartis Research Foundation (to RF), the European Research Council (ERC) under the European Union's Horizon 2020 research and innovation program (grant agreement No 742576 to RF), a fellowship from the Boehringer Ingelheim Fonds (to PR), and the UZH Alumni Research Talent Development fund (to ID). The funding bodies had no role in study design, data collection and interpretation, or the decision to submit the work for publication.

## Additional information

### Funding

| Funder | Grant reference number | Author |
| --- | --- | --- |
| Schweizerischer Nationalfonds zur Förderung der Wissenschaftlichen Forschung | PZ00P3_174018 | Igor Delvendahl |
| Schweizerischer Nationalfonds zur Förderung der Wissenschaftlichen Forschung | PZ00P3_209114 | Peter Rupprecht |
| Schweizerischer Nationalfonds zur Förderung der Wissenschaftlichen Forschung | 310030B_152833/1 | Rainer W Friedrich |
| National Institutes of Health | 1UM1MH136462 | Michael Z Lin |
| Novartis Foundation | | Rainer W Friedrich |
| European Research Council | 742576 | Rainer W Friedrich |
| Boehringer Ingelheim Fonds | | Peter Rupprecht |
| Deutsche Forschungsgemeinschaft | 535029399 | Igor Delvendahl |
| Deutsche Forschungsgemeinschaft | 535030493 | Igor Delvendahl |
| UZH Alumni Research Talent Development fund | | Igor Delvendahl |

The funders had no role in study design, data collection and interpretation, or the decision to submit the work for publication.

### Author contributions

Philipp S O'Neill, Conceptualization, Data curation, Software, Formal analysis, Investigation, Visualization, Writing – review and editing; Martín Baccino-Calace, Data curation, Software, Formal analysis, Investigation, Visualization, Writing – review and editing; Peter Rupprecht, Resources, Formal analysis,

Visualization, Writing – review and editing; Sungmoo Lee, Resources, Data curation, Writing – review and editing; Yukun A Hao, Resources, Writing – review and editing; Michael Z Lin, Rainer W Friedrich, Resources, Supervision, Writing – review and editing; Martin Mueller, Supervision, Funding acquisition, Writing – review and editing; Igor Delvendahl, Conceptualization, Resources, Data curation, Software, Formal analysis, Supervision, Funding acquisition, Investigation, Visualization, Writing – original draft, Project administration, Writing – review and editing

### Author ORCIDs
Philipp S O'Neill [iD] https://orcid.org/0009-0001-9208-7304
Peter Rupprecht [iD] https://orcid.org/0000-0001-8235-8257
Michael Z Lin [iD] https://orcid.org/0000-0002-0492-1961
Rainer W Friedrich [iD] https://orcid.org/0000-0001-9107-0482
Martin Mueller [iD] https://orcid.org/0000-0003-1624-6761
Igor Delvendahl [iD] https://orcid.org/0000-0002-6151-2363

### Ethics
Animals were treated according to national and institutional guidelines. All experiments were approved by the Cantonal Veterinary Office of Zurich (approval number no. ZH206/2016 and ZH009/2020).

Reviewer #1 (Public review): https://doi.org/10.7554/eLife.98485.3.sa1
Reviewer #2 (Public review): https://doi.org/10.7554/eLife.98485.3.sa2
Author response https://doi.org/10.7554/eLife.98485.3.sa3

## Additional files

### Supplementary files
MDAR checklist

### Data availability
All data generated or analysed during this study are included in the manuscript and supporting files. Datasets used for model training are available from Zenodo (https://doi.org/10.5281/zenodo.14507343). miniML source code and pre-trained models are available online (https://github.com/delvendahl/miniML, copy archived at *Delvendahl, 2025*) including analysis code. We used datasets from previous publications to generate training sets and to assess the application of miniML in zebrafish and *Drosophila* (*Rupprecht and Friedrich, 2018*; *Delvendahl et al., 2019*; *Baccino-Calace et al., 2022*). In addition, a published dataset (*Aggarwal et al., 2023*) was used to probe the application of miniML to glutamate imaging data (https://doi.org/10.25378/janelia.21985406).

The following previously published datasets were used:

| Author(s) | Year | Dataset title | Dataset URL | Database and Identifier |
|---|---|---|---|---|
| Podgorski K | 2023 | iGluSnFR3 Code and Data Supplement | https://doi.org/10.25378/janelia.21985406 | figshare, 10.25378/janelia.21985406 |
| O'Neill PS, Baccino Calace M, Rupprecht P, Lee S, Hao YA, Lin MZ, Friedrich RW, Mueller M, Delvendahl I | 2024 | Training data for miniML | https://doi.org/10.5281/zenodo.14507343 | Zenodo, 10.5281/zenodo.14507343 |

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
