## [Editor Report · eLife Assessment]

This paper presents miniML, an AI-based framework for the detection of synaptic events. Benchmark results presented in the paper are **compelling**, demonstrating the superiority of miniML over current state-of-the-art alternatives. The performance of miniML is demonstrated across various experimental paradigms, showing that miniML has the potential to become a **valuable** tool for the analysis of synaptic signals.

---

## [Referee Report · Reviewer #1 (Public review)]

O'Neill et al. have developed a software analysis application, miniML, that enables the quantification of electrophysiological events. They utilize a supervised deep learned-based method to optimize the software. miniML is able to quantify and standardize the analyses of miniature events, using both voltage and current clamp electrophysiology, as well as optically driven events using iGluSnFR3, in a variety of preparations, including in the cerebellum, calyx of held, golgi cell, human iPSC cultures, zebrafish, and *Drosophila*. The software appears to be flexible, in that users are able to hone and adapt the software to new preparations and events. Importantly, miniML is an open source software free for researchers to use and enables users to adapt new features using Python.

Overall this new software has the potential to become widely used in the field and an asset to researchers. Importantly, a new graphical user interface has been generated that enables more user control and a more user-friendly experience. Further, the authors demonstrate how miniML performs relative to other platforms that have been developed, and highlight areas where miniML works optimally. With these revisions, miniML should now be of considerable benefit and utility to a variety of researchers.

---

## [Referee Report · Reviewer #2 (Public review)]

Summary:

This paper presents miniML as a supervised method for detection of spontaneous synaptic events. Recordings of such events are typically of low SNR, where state-of-the-art methods are prone to high false favourable rates. Unlike current methods, training miniML requires neither prior knowledge of the kinetics of events nor the tuning of parameters/thresholds.

The proposed method comprises four convolutional networks, followed by a bi-directional LSTM and a final fully connected layer, which outputs a decision event/no event per time window. A sliding window is used when applying miniML to a temporal signal, followed by an additional estimation of events' time stamps. miniML outperforms current methods for simulated events superimposed on real data (with no events) and presents compelling results for real data across experimental paradigms and species.

Strengths:

The authors present a pipeline for benchmarking based on simulated events superimposed on real data (with no events). Compared to five other state-of-the-art methods, miniML leads to the highest detection rates and is most robust to specific choices of threshold values for fast or slow kinetics. A major strength of miniML is the ability to use it for different datasets. For this purpose, the CNN part of the model is held fixed and the subsequent networks are trained to adapt to the new data. This Transfer Learning (TL) strategy reduces computation time significantly and more importantly, it allows for using a substantially smaller data set (compared to training a full model) which is crucial as training is supervised (i.e. uses labeled examples).

Weaknesses:

The authors do not indicate how the specific configuration of miniML was set, i.e. number of CNNs, units, LSTM, etc. Please provide further information regarding these design choices, whether they were based on similar models or if chosen based on performance.

The data for the benchmark system was augmented with equal amounts of segments with/without events. Data augmentation was undoubtedly crucial for successful training.

(1) Does a balanced dataset reflect the natural occurrence of events in real data? Could the authors provide more information regarding this matter?

(2) Please provide a more detailed description of this process as it would serve users aiming to use this method for other sub-fields.

The benchmarking pipeline is indeed valuable and the results are compelling. However, the authors do not provide comparative results for miniML for real data (figures 4-8). TL does not apply to the other methods. In my opinion, presenting the performance of other methods, trained using the smaller dataset would be convincing of the modularity and applicability of the proposed approach.

Impact:

Accurate detection of synaptic events is crucial for the study of neural function. miniML has a great potential to become a valuable tool for this purpose as it yields highly accurate detection rates, it is robust, and is relatively easily adaptable to different experimental setups.

Comments on revisions:

The revised manuscript presents a compelling framework. The performance of mini ML is thouroughly explored and compared to several benchmarks. The training process along with other technical issues are now described in a satisfactory level of detail.

I think the authors did a great job. They answered all claims and concerns raised by me and the other reviewers.

---

## [Author Response]

The following is the authors’ response to the original reviews.

**Public Reviews:**

**Reviewer 1 (Public Review):**
O’Neill et al. have developed a software analysis application, miniML, that enables the quantification of electrophysiological events. They utilize a supervised deep learned-based method to optimize the software. miniML is able to quantify and standardize the analyses of miniature events, using both voltage and current clamp electrophysiology, as well as optically driven events using iGluSnFR3, in a variety of preparations, including in the cerebellum, calyx of held, Golgi cell, human iPSC cultures, zebrafish, and *Drosophila*. The software appears to be flexible, in that users are able to hone and adapt the software to new preparations and events. Importantly, miniML is an open-source software free for researchers to use and enables users to adapt new features using Python.Overall this new software has the potential to become widely used in the field and an asset to researchers. However, the authors fail to discuss or even cite a similar analysis tool recently developed (SimplyFire), and determine how miniML performs relative to this platform. There are a handful of additional suggestions to make miniML more user-friendly, and of broad utility to a variety of researchers, as well as some suggestions to further validate and strengthen areas of the manuscript:(1) miniML relative to existing analysis methods: There is a major omission in this study, in that a similar open source, Python-based software package for event detection of synaptic events appears to be completely ignored. Earlier this year, another group published SimplyFire in eNeuro (Mori et al., 2024; doi: 10.1523/eneuro.0326-23.2023). Obviously, this previous study needs to be discussed and ideally compared to miniML to determine if SimplyFire is superior or similar in utility, and to underscore differences in approach and accuracy.

We thank the reviewer for bringing this interesting publication to our attention. We have included SimplyFire in our benchmarking for comprehensive comparison with miniML. The approach taken by SimplyFire differs from miniML in a number of ways. Our results show that miniML provides higher recall and precision than SimplyFire (revised Figure 3). We appreciate that SimplyFire provides a user-interface similar to the commonly used MiniAnalysis software. In addition, the peak-finding-based approach of SimplyFire makes it relatively robust to event shape, which facilitates analysis of diverse data. However, we noted a strong threshold-dependence and long run time of SimplyFire (revised Figure 3 and Figure 3—figure supplement 1). In addition, SimplyFire is not robust against various types of noise typically encountered in electrophysiological recordings. Our extended benchmark analysis thus indicates that AI-based event detection is superior to existing algorithmic approaches, including SimplyFire.

(2) The manuscript should comment on whether miniML works equally well to quantify current clamp events (voltage; e.g. EPSP/mEPSPs) compared to voltage clamp (currents, EPSC/mEPSCs), which the manuscript highlights. Are rise and decay time constants calculated for each event similarly?

miniML works equally well for current- and voltage events (Figure 5, Figure 9). In general, events of opposite polarity can be analyzed by simply inverting the data. Transfer learning models may further improve the detection.

For each detected event, independent of data/recording type, rise times are calculated as 10–90% times (baseline–peak), and decay times are calculated as time to 50% of the peak. In addition, event decay time constants are calculated from a fit to the event average. With miniML being open-source, researchers can adapt the calculations of event statistics to their needs, if desired. In the revised manuscript, we have expanded the Methods section that describes the quantification of event statistics (Methods, Quantification).

(3) The interface and capabilities of miniML appear quite similar to Mini Analysis, the free software that many in the field currently use. While the ability and flexibility for users to adapt and adjust miniML for their own uses/needs using Python programming is a clear potential advantage, can the authors comment, or better yet, demonstrate, whether there is any advantage for researchers to use miniML over Mini Analysis or SimplyFire if they just need the standard analyses?

Following the reviewer’s suggestion, we developed a graphical user interface (GUI) for miniML to enhance its usability (Figure 2—figure supplement 2), which is provided on the GitHub repository. Our comprehensive benchmark analysis demonstrated that miniML outperforms existing tools such as MiniAnalysis and SimplyFire. The main advantages are (i) increased reliability of results, which eliminates the need for visual inspection; (ii) fast runtime and easy automation; (iii) superior detection performance as demonstrated by higher recall in both synthetic and real data; (iv) open-source Python-based design. We believe that these advantages make miniML a valuable tool for researchers recording various types of synaptic events, offering a more efficient and reliable solution compared to existing methods.

(4) Additional utilities for miniML: The authors show miniML can quantify miniature electrophysiological events both current and voltage clamp, as well as optical glutamate transients using iGluSnFR. As the authors mention in the discussion, the same approach could, in principle, be used to quantify evoked (EPSC/EPSP) events using electrophysiology, Ca2+ events (using GCaMP), and AP waveforms using voltage indicators like ASAP4. While I don’t think it is reasonable to ask the authors to generate any new experimental data, it would be great to see how miniML performs when analysing data from these approaches, particularly to quantify evoked synaptic events and/or Ca2+ (ideally postsynaptic Ca2+ signals from miniature events, as the *Drosophila* NMJ have developed nice approaches).

In the revised manuscript, we have extended the application examples of miniML. We applied miniML to detect mEPSPs recorded with the novel voltage-sensitive indicator ASAP5 (Figure 9 and Figure 9—figure supplement 1). We performed simultaneous recordings of membrane voltage through electrophysiology and ASAP5 voltage imaging in rat cultured neurons at physiological temperature. Data were analyzed using miniML, with electrophysiology data being used as ground-truth for assessing detection performance in imaging data. Our results demonstrate that miniML robustly detects mEPSPs in current-clamp, and can localize corresponding transients in imaging data. Furthermore, we observed that miniML performs better than template matching and deconvolution on ASAP5 imaging data (Figure 9 and Figure 9—figure supplement 2).

**Reviewer 2 (Public Review):**
This paper presents miniML as a supervised method for the detection of spontaneous synaptic events. Recordings of such events are typically of low SNR, where state-of-the-art methods are prone to high false positive rates. Unlike current methods, training miniML requires neither prior knowledge of the kinetics of events nor the tuning of parameters/thresholds.The proposed method comprises four convolutional networks, followed by a bi-directional LSTM and a final fully connected layer which outputs a decision event/no event per time window. A sliding window is used when applying miniML to a temporal signal, followed by an additional estimation of events’ time stamps. miniML outperforms current methods for simulated events superimposed on real data (with no events) and presents compelling results for real data across experimental paradigms and species. Strengths:The authors present a pipeline for benchmarking based on simulated events superimposed on real data (with no events). Compared to five other state-of-the-art methods, miniML leads to the highest detection rates and is most robust to specific choices of threshold values for fast or slow kinetics. A major strength of miniML is the ability to use it for different datasets. For this purpose, the CNN part of the model is held fixed and the subsequent networks are trained to adapt to the new data. This Transfer Learning (TL) strategy reduces computation time significantly and more importantly, it allows for using a substantially smaller data set (compared to training a full model) which is crucial as training is supervised (i.e. uses labeled examples).Weaknesses:The authors do not indicate how the specific configuration of miniML was set, i.e. number of CNNs, units, LSTM, etc. Please provide further information regarding these design choices, whether they were based on similar models or if chosen based on performance.The data for the benchmark system was augmented with equal amounts of segments with/without events. Data augmentation was undoubtedly crucial for successful training.(1) Does a balanced dataset reflect the natural occurrence of events in real data? Could the authors provide more information regarding this matter?

In a given recording, the event frequency determines the ratio of event-containing vs. nonevent-containing data segments. Whereas many synapses have a skew towards non-events, high event frequencies as observed, e.g., in pyramidal cells or Purkinje neurons, can shift the ratio towards event-containing data.

For model training, we extracted data segments from mEPSC recordings in cerebellar granule cells, which have a low mEPSC frequency (about 0.2 Hz, Delvendahl et al. 2019). Unbalanced training data may complicate model training (Drummond and Holte 2003; Prati et al. 2009; Tyagi and Mittal 2020). We therefore decided to balance the training dataset for miniML by down-sampling the majority class (i.e., non-event segments), so that the final datasets for model training contained roughly equal amounts of events and non-events.

(2) Please provide a more detailed description of this process as it would serve users aiming to use this method for other sub-fields.

We thank the reviewer for raising this point. In the revised manuscript, we present a systematic analysis of the impact of imbalanced training data on model training (Figure 1—figure supplement 2). In addition, we have revised the description of model training and data augmentation in the Methods section (Methods, Training data and annotation).

The benchmarking pipeline is indeed valuable and the results are compelling. However, the authors do not provide comparative results for miniML for real data (Figures 4-8). TL does not apply to the other methods. In my opinion, presenting the performance of other methods, trained using the smaller dataset would be convincing of the modularity and applicability of the proposed approach.

Quantitative comparison of synaptic detection methods on real-world data is challenging because the lack of ground-truth data prevents robust, quantitative analyses. Nevertheless, we compared miniML to common template-based and finite-threshold based methods on four different types of synapses. We noted that miniML generally detects more events, whereas other methods are susceptible to false-positives (Figure 4—figure supplement 1). In addition, we analyzed the performance of miniML on voltage imaging data (Figure 9). Simultaneous recordings of electrophysiological and imaging data allowed a quantitative comparison of detection methods in this dataset. Our results demonstrate that miniML provides higher recall for optical minis recorded using ASAP5 (Figure 9 and Figure 9—figure supplement 2; F1 score, Cohen’s d 1.35 vs. template matching and 5.1 vs. deconvolution).

Impact:Accurate detection of synaptic events is crucial for the study of neural function. miniML has a great potential to become a valuable tool for this purpose as it yields highly accurate detection rates, it is robust, and is relatively easily adaptable to different experimental setups.Additional comments:Line 73: the authors describe miniML as "parameter-free". Indeed, miniML does not require the selection of pulse shape, rise/fall time, or tuning of a threshold value. Still, I would not call it "parameter-free" as there are many parameters to tune, starting with the number of CNNs, and number of units through the parameters of the NNs. A more accurate description would be that as an AI-based method, the parameters of miniML are learned via training rather than tuned by the user.

We agree that a deep learning model is not parameter-free, and this term may be misleading. We have therefore changed this sentence in the introduction as follows: "The method is fast, robust to threshold choice, and generalizable across diverse data types [...]"

Line 302: the authors describe miniML as "threshold-independent". The output trace of the model has an extremely high SNR so a threshold of 0.5 typically works. Since a threshold is needed to determine the time stamps of events, I think a better description would be "robust to threshold choice".

To detect event localizations, a peak search is performed on the model output, which uses a minimum peak height parameter (or threshold). Extreme values for this parameter do indeed have a small impact on detection performance (Figure 3J). We have changed the description in the introduction and discussion according to the reviewer’s suggestion.

**Reviewer 3 (Public Review):**
miniML as a novel supervised deep learning-based method for detecting and analyzing spontaneous synaptic events. The authors demonstrate the advantages of using their methods in comparison with previous approaches. The possibility to train the architecture on different tasks using transfer learning approaches is also an added value of the work. There are some technical aspects that would be worth clarifying in the manuscript:(1) LSTM Layer Justification: Please provide a detailed explanation for the inclusion of the LSTM layer in the miniML architecture. What specific benefits does the LSTM layer offer in the context of synaptic event detection?

Our model design choice was inspired by similar approaches in the literature (Donahue et al. 2017; Islam et al. 2020; Passricha and Aggarwal 2019; Tasdelen and Sen 2021; Wang et al. 2020). Convolutional and recurrent neural networks are often combined for time-series classification problems as they allow learning spatial and temporal features, respectively. Combining the strengths of both network architectures can thus help improve the classification performance. Indeed, a CNN-LSTM architecture proved to be superior in both training accuracy and detection performance (Figure 1—figure supplement 2). Further, this architecture requires fewer free parameters than comparable model designs using fully connected layers instead. The revised manuscript shows a comparison of different model architectures (Figure 1—figure supplement 2), and we added the following description to the text (Methods, *Deep learning model architecture*):

"The combination of convolutional and recurrent neural network layers helps to improve the classification performance for time-series data. In particular, LSTM layers allow learning temporal features."

(2) Temporal Resolution: Can you elaborate on the reasons behind the lower temporal resolution of the output? Understanding whether this is due to specific design choices in the model, data preprocessing, or post-processing will clarify the nature of this limitation and its impact on the analysis.

When running inference on a continuous recording, we choose to use a sliding window approach with stride. Therefore, the model output has a lower temporal resolution than the raw data, which is determined by the stride length (i.e., how many samples to advance the sliding window). While using a stride is not required, it significantly reduces inference time (cf. Figure 2—figure supplement 1). We recommend a stride of 20 samples, which does not impact the detection of events. Any subsequent quantification of events (amplitude, area, risetimes, etc.) is performed on raw data. Based on the reviewer’s comment, we have adapted the code to resample the prediction trace to the sampling rate of the original data. This maintains temporal precision and avoids confusion.

The Methods now include the following statement:

"To maintain temporal precision, the prediction trace is resampled to the sampling frequency of the raw data."

(3) Architecture optimization: how was the architecture CNN+LSTM optimized in terms of a number of CNN layers and size?

We performed a Bayesian optimization over a defined range of hyperparameters in combination with empirical hyperparameter tuning. We now describe this in the Methods section as follows:

"To optimise the model architecture, we performed a Bayesian optimisation of hyperparameters. Hyperparameter ranges were chosen for the free parameters of all layers. Optimisation was then performed with a maximum number of trials of 50. Models were evaluated using the validation dataset. Because higher number of free parameters tended to increase inference times, we then empirically tuned the chosen hyperparameter combination to achieve a trade-off between number of free parameters and accuracy."

**Recommendations For The Authors**

**Reviewing Editor (Recommendations For The Authors):**
Overall suggestions to the authors:(1) Directly compare miniML with SimplyFire (which was not cited or discussed in the original manuscript), with both idealized and actual data. Discuss the pros/cons of each software.

We have conducted an extensive comparison between miniML and SimplyFire using both simulated and actual experimental data. This analysis is now presented in the revised Figure 3, Figure 3—figure supplement 1, and Figure 4—figure supplement 1. In addition, we have included relevant citations for SimplyFire in our manuscript. These additions provide a more comprehensive and balanced view of the available tools in the field, positioning our work within the broader context of existing solutions.

(2) Generate a better user interface akin to MiniAnalysis or SimplyFire.

We thank the editor and reviewers for the suggestion to improve the user interface. We have created a user-friendly graphical user interface (GUI) for miniML that is available on our GitHub repository. This GUI is now showcased in Figure 2—figure supplement 2 of the manuscript. The new interface allows users to load and analyze data through an intuitive point-and-click system, visualize results in real-time, and adjust parameters easily without coding knowledge. We have incorporated user feedback to refine the interface and improve user experience. These improvements significantly enhance the accessibility of miniML, making it more user-friendly for researchers with varying levels of programming expertise.

**Reviewer 1 (Recommendations For The Authors):**
Related to point (1) of the Public Review, we have taken the liberty to compare electrophysiological data using miniAnalysis, SimiplyFire, and miniML. In our comparison, we note the following in our experience:(1.1) In contrast to both SimplyFire and miniAnalysis, miniML does not currently have a user-friendly interface where the user can directly control or change the parameters of interest, nor does miniML have a user control center, so the user cannot simply type or select the mini manually. Rather, if any parameter needs to be changed, the user needs to read, understand, and change the original source code to generate the preferred change. This level of "activation energy" and required user coding expertise in computer science, which many researchers do not have, renders miniML much less accessible when directly compared to SimplyFire and miniAnalysis. Hence, unless miniML’s interface can be made more user-friendly, this is a major disadvantage, especially when compared to SimplyFire, which has many of the same features as miniML but with a much easier interface and user controls.

As suggested by the reviewer, we have created a graphical user interface (GUI) for miniML. The GUI allows easy data loading, filtering, analysis, event inspection, and saving of results without the need for writing Python code. Figure 2—figure supplement 2 illustrates the typical workflow for event analysis with miniML using the GUI and a screenshot of the user interface. Code to use miniML via the GUI is now included in the project’s GitHub repository. The GUI provides a simple and intuitive way to analyze synaptic events, whereas running miniML as Python script allows for more customization and a high degree of automatization.

(1.2) We compared electrophysiological miniature events between miniML, SimplyFire, and miniAnalysis. All three achieved similar mean amplitudes in "wild type" conditions, and conditions in which mini events were enhanced and diminished, so the overall means and utilities are similar, with miniML and SimplyFire being preferred given the flexibility and much faster analysis. We did note a few differences, however. SimplyFire tends to capture a high number of mini-events over miniML, especially in conditions of diminished mini amplitude (e.g., miniML found 76 events, while SimplyFire 587). The mean amplitudes, however, were similar. It seems that in data with low SNR, SimplyFire captures many more events as real minis that are probably noise, while miniML is more selective, which might be an advantage in miniML. That being said, we found SimplyFire to be superior in many respects, not least of which the user interface and experience.

We appreciate the reviewer’s thorough comparison of miniML, SimplyFire, and MiniAnalysis. While we acknowledge SimplyFire’s user-friendly interface, our study highlights several advantages of AI-based event analysis over conventional algorithmic approaches. Our updated benchmark analysis revealed better detection performance of miniML compared with SimplyFire (revised Figure 3), which had similar performance to deconvolution. As already noted by the reviewer, high false positive rates are a major issue of the SimplyFire approach. Although a minimum amplitude cutoff can partially resolve this problem, detection performance is highly sensitive to threshold setting (revised Figure 3). Another apparent disadvantage of SimplyFire is its relatively slow runtime (Figure 3—figure supplement 1). Finally, we have enhanced miniML’s accessibility by providing a graphical user interface that is easy to use and provides additional functionality.

Some technical comments:(1) Improvements to the dependence version of miniML: There is a need to clarify the dependence version of the python and tensor flow used in this study and in the GitHub. We used Python version 3.8.19 to load the miniML model. However, if Python versions > = 3.9, as described on the GitHub provided, it is difficult to have a matched h5py version installed. It is also inaccurate to say using Python > = 3.9, because tensor flow version for this framework needs to be around 2.13. However, if using Python > = 3.10, it will only allow 2.16 version tensor flow to be the download choice. Therefore, as a Python framework, the dependency version needs to be specified on GitHub to allow researchers to access the model using the entire work.

Thank you for highlighting this issue. We have now included specific version numbers in the requirements to avoid version conflicts and to ensure proper functioning of the code.

(2) Due to the intrinsic characteristics of the trained model, every model is only suitable for analyzing data with similar attributes. It is hard for researchers without a strong computer science background to train a new model themselves for their specific data. Therefore, it would be preferred if there were more available transfer learning models on GitHub accessible for researchers to adapt to their data.

We would like to thank the reviewer for this feedback. Trained models (such as the default model) can often be used on different data (see, e.g., Figure 4, where data from four distinct synaptic preparations were analyzed with the base model, and Figure 5—figure supplement 1). However, changes in event waveform and/or noise characteristics may necessitate transfer learning to obtain optimal results with miniML. We have revised the description and tutorial for model training on the project’s GitHub repository to provide more guidance in this process. In addition, we now provide a tutorial on how to use existing models on out-of-sample data with distinct kinetics, using resampling. We hope these updates to the miniML GitHub repository will facilitate the use of the method.

Following the suggestion by the reviewer, we have provided the transfer learning models used for the manuscript on the project’s GitHub repository to increase the number of available machine learning models for event detection. In addition, users of miniML are encouraged to supply their custom models. We hope that this will facilitate model exchange between laboratories in the future.

**Reviewer 3:**
I congratulate all authors for the convincing demonstration of their methodology, I do not have additional recommendations.

We would like to thank the reviewer for the positive assessment of our manuscript.

References

Delvendahl, I., Kita, K., & Müller, M. (2019). Rapid and sustained homeostatic control of presynaptic exocytosis at a central synapse. *Proceedings of the National Academy of Sciences*, *116*(47), 23783–23789. https://doi.org/10.1073/pnas.1909675116

Donahue, J., Hendricks, L. A., Rohrbach, M., Venugopalan, S., Guadarrama, S., Saenko, K., & Darrell, T. (2017). Long-term recurrent convolutional networks for visual recognition and description. *IEEE Transactions on Pattern Analysis and Machine Intelligence*, *39*(4), 677–691. https://doi.org/10.1109/tpami.2016.2599174

Drummond, C., & Holte, R. C. (2003). C4.5, class imbalance, and cost sensitivity: Why under-sampling beats over-sampling. https://api.semanticscholar.org/CorpusID:204083391

Islam, M. Z., Islam, M. M., & Asraf, A. (2020). A combined deep CNN-LSTM network for the detection of novel coronavirus (COVID-19) using x-ray images. *Informatics in Medicine Unlocked*, *20*, 100412. https://doi.org/10.1016/j.imu.2020.100412

Passricha, V., & Aggarwal, R. K. (2019). A hybrid of deep CNN and bidirectional LSTM for automatic speech recognition. *Journal of Intelligent Systems*, *29*(1), 1261–1274. https://doi.org/10.1515/jisys-2018-0372

Prati, R. C., Batista, G. E. A. P. A., & Monard, M. C. (2009). Data mining with imbalanced class distributions: Concepts and methods. Indian International Conference on Artificial Intelligence. https://api.semanticscholar.org/CorpusID:16651273

Tasdelen, A., & Sen, B. (2021). A hybrid CNN-LSTM model for pre-miRNA classification. *Scientific Reports*, *11*(1). https://doi.org/10.1038/s41598-021-93656-0

Tyagi, S., & Mittal, S. (2020). Sampling approaches for imbalanced data classification problem in machine learning. In P. K. Singh, A. K. Kar, Y. Singh, M. H. Kolekar, & S. Tanwar (Eds.), *Proceedings of icric 2019* (pp. 209–221). Springer International Publishing.

Wang, H., Zhao, J., Li, J., Tian, L., Tu, P., Cao, T., An, Y., Wang, K., & Li, S. (2020). Wearable sensor-based human activity recognition using hybrid deep learning techniques. *Security and Communication Networks*, *2020*, 1–12. https://doi.org/10.1155/2020/2132138